# Vertically stacked, low-voltage organic ternary logic circuits including nonvolatile floating-gate memory transistors

Junhwan Choi[1,4], Changhyeon Lee[1,4], Chungryeol Lee[1], Hongkeun Park [1], Seung Min Lee[1], Chang-Hyun Kim [2], Hocheon Yoo [2✉] & Sung Gap Im [1,3✉]

Multi-valued logic (MVL) circuits based on heterojunction transistor (HTR) have emerged as an effective strategy for high-density information processing without increasing the circuit complexity. Herein, an organic ternary logic inverter (T-inverter) is demonstrated, where a nonvolatile floating-gate flash memory is employed to control the channel conductance systematically, thus realizing the stabilized T-inverter operation. The 3-dimensional (3D) T-inverter is fabricated in a vertically stacked form based on all-dry processes, which enables the high-density integration with high device uniformity. In the flash memory, ultrathin polymer dielectrics are utilized to reduce the programming/erasing voltage as well as operating voltage. With the optimum programming state, the 3D T-inverter fulfills all the important requirements such as full-swing operation, optimum intermediate logic value (~$V_{DD}$/2), high DC gain exceeding 20 V/V as well as low-voltage operation (< 5 V). The organic flash memory exhibits long retention characteristics (current change less than 10% after $10^4$ s), leading to the long-term stability of the 3D T-inverter. We believe the 3D T-inverter employing flash memory developed in this study can provide a useful insight to achieve high-performance MVL circuits.

[1] Department of Chemical and Biomolecular Engineering Korea Advanced Institute of Science and Technology (KAIST) 291 Daehak-ro, Yuseong-gu, Daejeon 34141, Korea. [2] Department of Electronic Engineering Gachon University 1342 Seongnam-daero, Sujeong-gu, Seongnam, Gyeonggi-do 13120, Korea. [3] KAIST Institute For NanoCentury (KINC) Korea Advanced Institute of Science and Technology (KAIST) 291 Daehak-ro, Yuseong-gu, Daejeon 34141, Korea. [4]These authors contributed equally: Junhwan Choi, Changhyeon Lee. ✉email: hyoo@gachon.ac.kr; sgim@kaist.ac.kr

Organic thin-film transistors (OTFTs) have gained tremendous attention as a building block for the next-generation electronic devices such as wearable/stretchable electronics[1–3], flexible sensors[4,5], and electronic skins[6,7]. With the numerous efforts in the last two decades, the performance of OTFT and OTFT-based integrated circuits (ICs) has been enhanced remarkably[8–11]. However, the strong susceptibility of organic materials to high temperature and organic solvent has significantly hampered their compatibility with lithography-based down-scaling[12,13]. Alternatively, three-dimensional (3D) vertical stacking of OTFTs has been demonstrated to increase the number of transistors per unit area, without reducing the device size[14–16]. Among the various approaches to fabricating vertically stacked devices, such as etching-based via-hole processes or inkjet printing[14,15,17], a via-hole-less metal interconnects scheme for OTFTs enabled by forming an insulating layer on a selective area was proposed as an attractive way to minimize damage to the underlying layers or devices[16,18].

In addition to the 3D stacking of the devices, multi-valued logic (MVL) devices have emerged as a promising strategy to accommodate the increasing demand for higher data processing capability in advanced electronic systems such as artificial intelligence and internet-of-things (IoT)[19–21]. When compared to conventional binary logic, which has two simple distinct logic states ("0" and "1"), MVL can process signals with fewer interconnects, allowing for a significant reduction in the number of devices and hence the complexity of the ICs[22,23]. For example, ternary logic can reduce the system complexity reportedly to ~63% compared to the binary logic in principle[24]. However, the conventional complementary metal-oxide-semiconductor (CMOS) technology requires at least four additional transistors to demonstrate ternary logic, which unavoidably results in increased complexity and power consumption[25,26]. To overcome these limitations, heterojunction transistors (HTRs) displaying negative differential resistance (NDR) or negative transconductance (NTC) have been proposed by utilizing various device designs and materials such as organic semiconductors[27–29], 2D materials[30–35], and organic–inorganic heterostructures[36–38]. The key advantage of the HTR lies in the fact that the ternary logic can be implemented simply by replacing a transistor with an HTR without increasing the number of devices to achieve the ternary logic.

Organic HTRs have been developed based on a partial junction between p-and n-type organic semiconductors in the active layer, using the intrinsic semiconducting feature of organic semiconductors that makes them suitable for achieving highly ordered heterointerfaces[20,28]. Such anti-ambipolar transistors (AATs) with NTC behavior displayed a "flipped V"-shaped transfer characteristic, which could create intermediate logic states[27]. Despite the successful demonstration of the ternary logic inverter (T-inverter), however, the full swing of the output voltage ($V_{OUT}$) was hardly achieved, mainly due to the inherently low on/off current ratio ($I_{on}/I_{off}$) of the AATs[27,39]. To address this issue, a modified HTR structure was proposed where one of the semiconducting layers was in contact with both source (S) and drain (D) electrodes (main-semiconductor), on top of the partial semiconducting layer with the opposite polarity (sub-semiconductor)[21,29]. In contrast with AATs, the main-semiconductor connected from S to D electrodes led to high on-current ($I_{on}$) at higher gate voltage ($V_G$) while maintaining the NTC characteristics in the modified HTR, which enabled the full-swing operation of the T-inverter[29]. However, it is still challenging to satisfy all the important requirements of T-inverters including (i) a well-defined intermediate logic state with the optimized value ($V_{OUT}$~half of the supply voltage ($V_{DD}/2$), logic "1") in the sufficient input voltage ($V_{IN}$) range, (ii) full-swing operation ($V_{OUT}$ from $V_{DD}$ (logic "2") to ground ($G_{ND}$) (logic "0")) as well as (iii) hysteresis-free, low-voltage operation. One of the most critical obstacles to achieving the optimum intermediate logic value is the mismatch in channel conductance between the HTR and complementary transistor, particularly in the NTC range. Therefore, it is imperative to develop a systematic strategy to optimize T-inverter and constituent transistor performance.

As an effective method to systematically control the electrical characteristics (i.e., channel conductance) of a transistor, a nonvolatile floating-gate (FG) flash memory was utilized and integrated with HTR to implement a T-inverter in this study. The implementation of the nonvolatile FG flash memory enabled systematic circuit programmability, where the precise control of the threshold voltage ($V_T$) allows for enhancement of the T-inverter performance by optimizing the intermediate logic state. Furthermore, by utilizing the via-hole-less metal interconnection method[16,18], the flash memory on the first floor was vertically integrated with the HTR on the second floor to realize a vertically stacked, 3D T-inverter. Considering the unit inverter logic, the vertical stacking effect can improve the information density by a factor of 2, and the ternary logic effect can enhance the information density by a factor of 1.5, so the proposed vertically stacked ternary circuit can further improve the information density by a factor of 3. Moreover, the integration density can increase exponentially as the circuit configuration becomes more complex. It is worthwhile to note that the introduction of flash memory is a powerful strategy in that the adjustability of the channel dimension is limited in the stacked structure.

In the 3D T-inverter fabrication process, a vapor-phase deposition method, termed initiated chemical vapor deposition (iCVD) process was introduced to deposit the high-performance polymer dielectric layers while minimizing the damage to the underlying devices[40–43]. The polymer dielectric materials were introduced with an optimized dielectric layer configuration, which reduced the programming/erasing voltage ($V_{prg}/V_{ers}$), and enabled the operating voltage <5 V, which is comparable to or even lower than those obtained in the previous organic 3D logic circuits[14–16]. Meanwhile, the high device-to-device uniformity was also fully retained. Most of all, the solvent-free deposition process was capable of in situ patterning of the dielectric layer simply through shadow mask during the deposition process, which allowed etching-free, via-hole-less metal interconnection, and thus 3D vertical integration of the HTR and flash memory devices[16]. According to the programming/erasing states of the flash memory device, the intermediate logic value of the 3D T-inverter could be controlled systematically. The 3D T-inverter with the optimized memory programming state exhibited long retention characteristics and operational stability against the repeated operation cycles.

## Results

**Design of the organic 3D T-inverter.** The schematic illustration for the fabrication process and the structure of the vertically stacked organic ternary logic inverter are shown in Fig. 1a, b, respectively. All the dielectric layers were deposited by iCVD process and patterned in situ through a shadow mask to achieve via-hole-less metal interconnection[16]. Also, a 1 μm-thick interlayer dielectric (ILD) was imposed between the unit devices to electrically isolate the flash memory and HTR. A schematic illustration of the device structure with various viewpoints according to the fabrication process is shown in Supplementary Fig. 1. To optimize the voltage transfer characteristic (VTC) and the intermediate logic state of the resulting 3D T-inverter, organic flash memory with FG was used to match the channel conductance with the HTR in the NTC area (Fig. 1c, d). The n-type

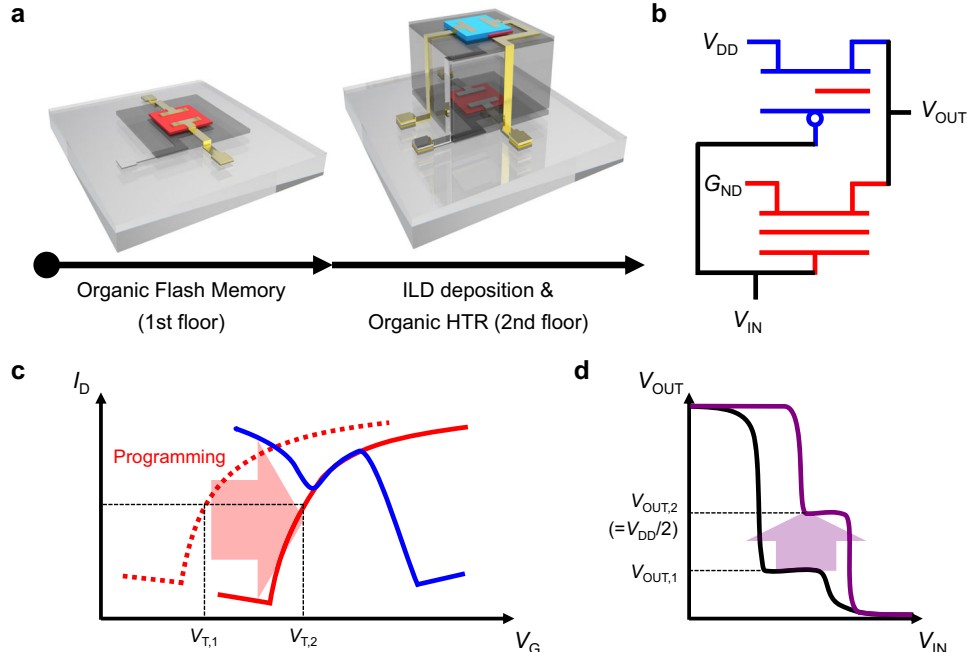

**Fig. 1 Design of 3D T-inverter including the flash memory. a** A schematic of device fabrication and **b** the vertically stacked 3-dimensional (3D) ternary logic inverter (T-inverter) consisting of the flash memory and heterojunction transistor (HTR). The schematic illustrations of **c** the transfer curve shift along with the memory programming, and **d** corresponding voltage transfer characteristic (VTC) of the 3D T-inverter.

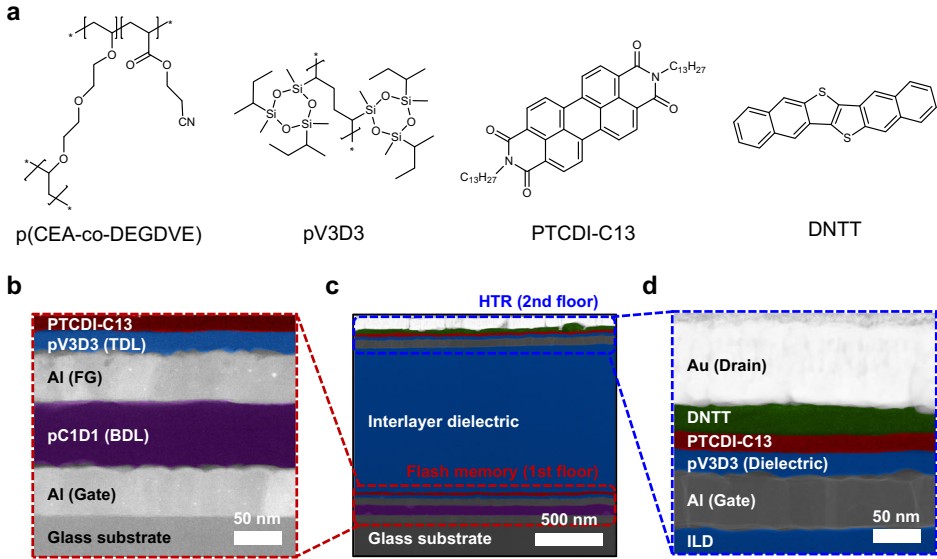

**Fig. 2 Materials and device structures of the 3D T-inverter. a** The chemical structures of the polymer dielectrics and organic semiconductors used in this study. The high-resolution transmission electron microscope (HRTEM) images of the **b** flash memory, **c** 3-dimensional (3D) ternary logic inverter (T-inverter), and **d** heterojunction transistor (HTR). The false-color modification was attempted to organic layers to distinguish each layer in HRTEM images.

flash memory and HTR were utilized as a pull-down and pull-up transistor and those devices were fabricated on the first and second floor, respectively. It should be noted that the drain electrode on top of the heterojunction in HTR was interconnected to the drain electrode in the flash memory and used as an output electrode in the inverter operation to induce an intermediate logic state[29]. The chemical structures of the polymer dielectrics and organic semiconductors used in this study are shown in Fig. 2a. For the polymer dielectric layers, poly(2-cyanoethyl acrylate-*co*-diethylene glycol divinyl ether) [p(CEA-*co*-DEGDVE)] (named pC1D1), whose composition was optimized to maximize the

dielectric constant (the dielectric constant, $k > 6$) while retaining the insulating performance (breakdown field, $E_{break} > 3$ MV/cm)[42], was used as blocking dielectric layer (BDL) in this study. Poly(1,3,5-trivinyl-1,3,5-trimethyl cyclotrisiloxane) (pV3D3) with a low dielectric constant ($k < 2.3$)[40] was also employed as a tunneling dielectric layer (TDL) in the flash memory, and gate dielectric layer (GDL) in the HTR to provide a non-polar interface for the facilitated charge transport in the organic semiconductors[44]. The dielectric layers were deposited using the iCVD process, which commonly exhibited robust insulating performance even at a thickness of 100 nm for BDL and 24 nm

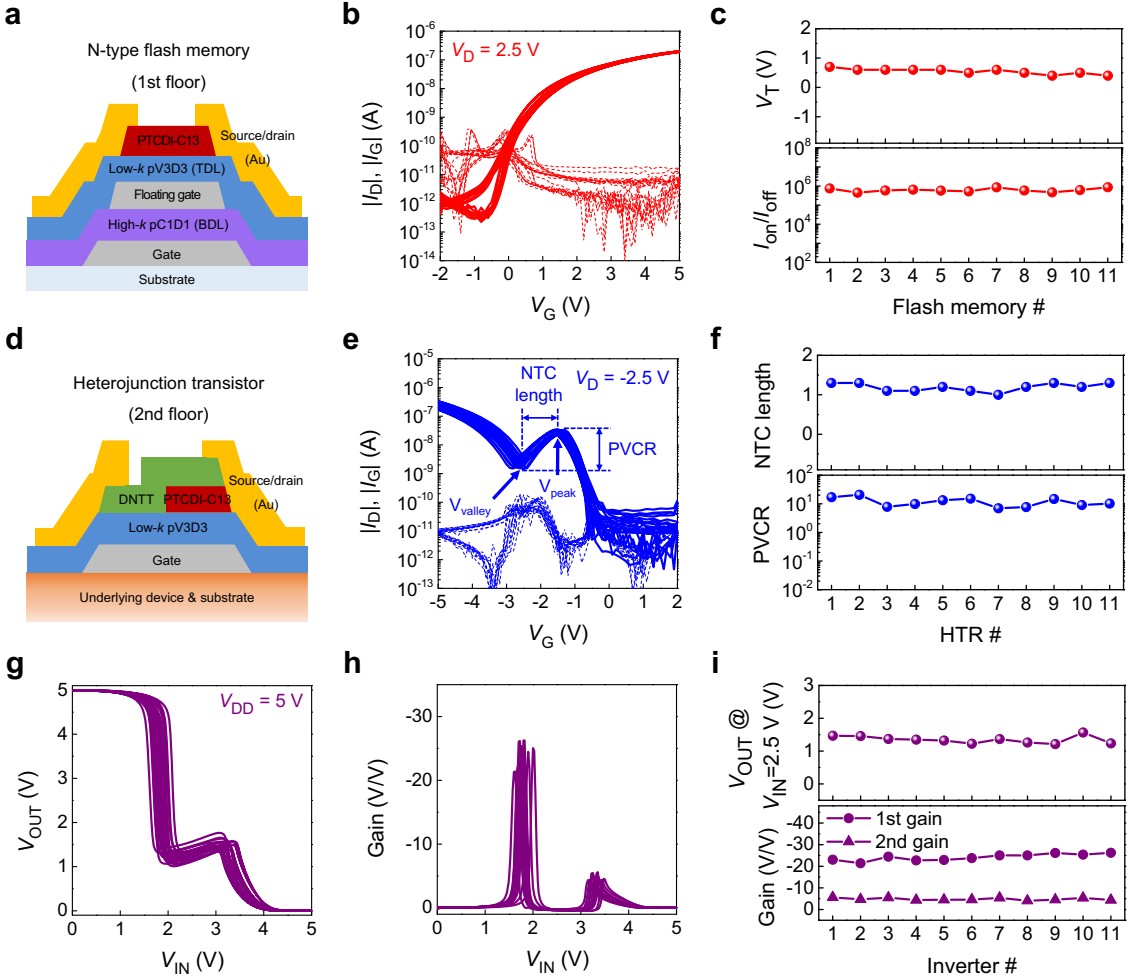

**Fig. 3 Device-to-device uniformity. a** Schematic illustration of the organic flash memory on the first floor, **b** transfer characteristics of the 11 memory devices, and **c** their threshold voltage ($V_T$), on/off current ratio ($I_{on}/I_{off}$) variation. **d** Schematic illustration of the heterojunction transistor (HTR) on the second floor, **e** transfer characteristics of the 11 HTR devices, and **f** their negative transconductance (NTC) length, peak-to-valley current ratio (PVCR) variation. **g** voltage transfer characteristics (VTCs), **h** DC gain profiles, and **i** intermediate logic value, DC gain variation of the 11 3-dimensional (3D) ternary logic inverter (T-inverter) devices.

for TDL (Supplementary Figs. 2 and 3). Dinaphtho[2,3-b:2',3'-f] thieno[3,2-b]thiophene (DNTT) and *N,N'*-ditridecylperylene-3,4,9,10-tetracarboxylic diimide (PTCDI-C13) were utilized as p- and n-type organic semiconductors, respectively, which show charge transport characteristics that are comparable to each other[29,45]. The high-resolution transmission electron microscope (HRTEM) images of the flash memory, 3D T-inverter and HTR are shown in Fig. 2b–d, respectively. The vertical integration of the device was achieved without damaging the underlying layers or devices as all the fabrication processes were based on solvent-free, dry methods with mild process conditions[16,40,42]. In addition, no notable defect or intermixing was observed in the energy dispersive spectroscopy (EDS) mapping analysis (Supplementary Fig. 4). These results clearly exhibited that the flash memory-integrated organic 3D T-inverter was successfully fabricated in the vertically stacked structure which can enhance the integration density by increasing the number of devices per unit area[14–16].

**Electrical characteristics of the devices**. Fig. 3a shows the schematic illustration of n-type PTCDI-C13 flash memory consisting of the FG sandwiched by high-*k* pC1D1 and low-*k* pV3D3 as BDL and TDL layers, respectively. In the operation of flash memory with FG, the electric field (*E*) applied to each TDL and

BDL ($E_{TDL}/E_{BDL}$) is scaled with the gate coupling ratio, $\alpha_{CR}$ ($=V_{FG}/V_G$, where $V_{FG}$ is the amount of voltage applied to FG) which is associated with the relative capacitance of the dielectric layers ($C_{BDL}/(C_{BDL} + C_{TDL})$), where $C$ is capacitance)[46,47]. In other words, the increased $C_{BDL}$ is important to distribute higher effective voltage to the TDL. In our flash memory device, the dielectric constant of the pC1D1 BDL (~6.2) is about 2.8 times higher than that of the pV3D3 TDL (~2.2). Therefore, far higher *E* is applied to TDL to facilitate Fowler-Nordheim (F-N)-like tunneling through pV3D3 TDL, while the *E* applied to BDL is quite small, which can efficiently prevent the charge leakage through BDL (Supplementary Fig. 3). The programming/erasing voltage ($V_{prg}/V_{ers}$) of flash memory might be significantly decreased by carefully designing the dielectric layer structure while taking into account the dielectric constant of each material. Moreover, employing high-*k* BDL also enabled low-voltage operation while retaining thicker dielectric layer (~100 nm), which is highly desirable for securing the reliable device operation. The cumulative transfer characteristics of the 11 flash memory devices are shown in Fig. 3b, where all the devices exhibited distinct transistor characteristics. The applied drain voltage ($V_D$) was 2.5 V for the flash memory and −2.5 V for the HTR. All the measured transfer curves were hysteresis-free, due to the extremely low leakage current through the TDL in the

given operating voltage range. As shown in the statistical analysis, all the flash memory devices exhibited on/off current ratio ($I_{on}$/$I_{off}$) higher than $10^5$ with the operating voltage <5 V (Fig. 3c). Furthermore, the average $V_T$ was <0.7 V with narrow distribution (0.55 ± 0.09 V), thus ensuring the uniform channel conductance of the pull-down transistor. It is also worthwhile to note that even with a relatively thick polymer ILD with low thermal conductivity, only negligible change in the temperature of the flash memory was observed during the continuous operation at 5 V, owing to the low operating voltage (Supplementary Figs. 5 and 6).

The device structure of the HTR is shown in Fig. 3d, where n-type PTCDI-C13 was deposited and annealed to obtain the enlarged grain with high crystallinity, followed by the deposition of p-type DNTT film. An atomic force microscope (AFM) analysis clearly verified the formation of the heterojunction structure at the middle edge of the heterojunction (Supplementary Fig. 7). As a GDL, pV3D3 was utilized, enabling the interfacial matching with both PTCDI-C13 and DNTT, as well as imparting thermal stability to survive the thermal annealing step at 200 °C[16,45]. The DNTT film connected the S and D electrodes, which allowed to induce hole accumulation at high |$V_G$|, thus leading to the high $I_{on}$/$I_{off}$[29]. The thickness of the GDL in the HTR was optimized to 50 nm, to match the capacitance per unit area ($C_i$) of the flash memory and HTR. All the HTR exhibited NTC characteristics with four different operation regions: (i) off-state (−0.8 V < $V_G$ < 2 V), (ii) subthreshold region (−1.5 V (=$V_{peak}$, peak voltage) < $V_G$ < −0.8 V) where both hole-induced current and electron-induced band-to-band tunneling (BTBT) contributed to drain current ($I_D$). (iii) NTC region (−2.6 V (=$V_{valley}$, valley voltage) < $V_G$ < −1.5 V) where a depletion of n-type PTCDI-C13 occurred, thus |$I_D$| was decreased with the increasing |$V_G$|, and (iv) on-state (−5 V < $V_G$ < −2.6 V) with more hole accumulation in the higher |$V_G$| range (Fig. 3e). A more detailed description of the origin of $I_D$ in each regime of HTR can be found in Supplementary Fig. 8. All the HTR devices exhibited distinct NTC behaviors in the transfer characteristic. The NTC length was 1.09 ± 0.10 V, and the peak-to-valley current ratio ($I_{peak}$/$I_{valley}$) was >12.0 in the NTC region, as defined by the $V_G$ range whereby |$I_D$| decreased with the increasing |$V_G$| (Fig. 3f). In addition, the narrow distribution of $V_T$, $V_{peak}$, and $V_{valley}$ was also confirmed in the HTR transfer curves (Supplementary Fig. 9).

The VTC and gain profile of the vertically integrated 3D T-inverter are shown in Fig. 3g, h, respectively. Through the successful operation of the 3D T-inverter, it was confirmed that the electrical interconnection of the electrodes between the top and bottom floors, separated by the shadow-mask patterned dielectric layers. All the fabricated 3D T-inverters showed full-swing operation with the uniform electrical characteristics stemming from the high device-to-device uniformity of the constituent devices, together with the high $I_{on}$/$I_{off}$ ratio of the HTR, which can be hardly achieved in conventional AATs where two semiconductors are overlapped only at the center of the channel[27,28]. Also, no notable hysteresis was observed in the transfer curves of the flash memory and HTR, and the VTC of the T-inverter regardless of the sweeping speed (Supplementary Figs. 10 and 11). Moreover, all the 3D T-inverters displayed a distinct intermediate logic state with two clearly distinguishable maximum gain values. The maximum DC gain in the first peak (1st gain) value was higher than the maximum DC gain in the second peak (2nd gain) value due to the relatively low intermediate logic value compared to the $V_{DD}$/2. The intermediate logic value was located not in the middle value, but leaned toward 0 V, indicating that the channel conductance of the pull-down transistor (flash memory) was higher than that of the pull-up transistor (HTR), which can be further tuned exquisitely

by adjusting the channel conductance of the flash memory. The statistical analysis revealed that the high device-to-device uniformity of the 3D T-inverter with a narrow distribution of the intermediate logic value ($V_{OUT}$~1.35 ± 0.11 V at $V_{IN}$ = 2.5 V) and maximum DC gain values (−24.21 ± 1.55 and −4.81 ± 0.54 V/V for the 1st and 2nd gain, respectively), even with the 3D stacked structure (Fig. 3i).

**Optimization of the 3D T-inverter characteristics.** Positive (programming) or negative (erasing) gate biases were applied to the gate (G) electrode to pump electrons from the channel to the FG or to detrap the stored charges from the FG, respectively, to analyze memory characteristics (Supplementary Fig. 12). The memory programming and erasing window was investigated via an incremental step pulse programming (ISPP) and erasing (ISPE), respectively, and the pulse width was commonly set to 1 s (Fig. 4a). The change in transfer curves of the memory device in ISPP operation is shown in Fig. 4b. The transfer curve was shifted gradually toward the positive direction with the increasing $V_{prg}$ without any notable degradation of the transistor characteristics nor apparent hysteresis behavior. The critical $V_G$ ($V_C$), where $V_T$ shift (Δ$V_T$) starts to occur was as low as 12 V thanks to the high $α_{CR}$. The Δ$V_T$ also increased gradually with the increasing $V_{prg}$, and reached 2.5 V with a maximum $V_{prg}$ of +19 V, which was quite large considering the operating voltage (<5 V) (Fig. 4c). Based on the observed programming operation, it was verified that the n-type channel conductance can be controlled systematically by adjusting $V_{prg}$. In addition, the $V_T$ of the memory device could also be controlled by applying $V_{prg}$ or $V_{ers}$ with various Δt and the $V_T$ split was enlarged in accordance with the increasing Δt (Supplementary Fig. 13). We also investigated the cycling endurance where the memory device exhibited reversible and reliable programming/erasing operation with $I_D$ ratio higher than $10^3$ (at $V_G$ = 1 V) over 50 cycles (Supplementary Fig. 14).

The electrical characteristics of the vertically stacked 3D T-inverter were analyzed according to the programming/erasing states of the flash memory on the first floor (Fig. 4d, e). The $I_D$ of the flash memory was modulated successfully by programming operation (Supplementary Fig. 15), and with the decreasing channel conductance by charge trapping into the FG, the intermediate logic state was systematically controlled as shown in the VTC of the 3D T-inverter (Fig. 4f). The intermediate logic value was gradually increased in accordance with the increasing $V_{prg}$ and reached 2.49 V with $V_{prg}$ = + 19 V (Fig. 4g). As a result, the optimal programming state of the flash memory allowed the intermediate logic value to be practically identical to $V_{DD}$/2 (logic "1"), which made the intermediate logic state clearly distinguishable from $V_{DD}$ (logic "2") and $G_{ND}$ (logic "0"). The $V_{IN}$ range where the intermediate logic states appeared was gradually shifted with the increasing $V_{prg}$, because the NTC region of the HTR was shifted to lower |$V_G$| region with the decreasing |$V_D$| (Supplementary Fig. 16). The $V_{IN}$ range of the intermediate logic states decreased slightly with the increasing $V_{prg}$, because only the NTC region in the HTR could be represented in the intermediate logic state, which enabled us to obtain the optimum $V_{out}$ value ($V_{DD}$/2) by optimizing the memory programming process (Supplementary Fig. 17). The 1st gain decreased gradually while 2nd gain increased with the increasing $V_{prg}$, resulting from the increased intermediate logic value (Fig. 4h and Supplementary Fig. 18). Nevertheless, the obtained gain value was still high, compared to the previously reported non-Si-based ternary logic inverters reported to date (Supplementary Table 1)[23,29,30,32–34,37]. Also, the obtained gain values were comparable to those observed in the organic binary logic inverters with polymer dielectric materials[48–56]. Moreover, the dramatic increase of the 2nd gain

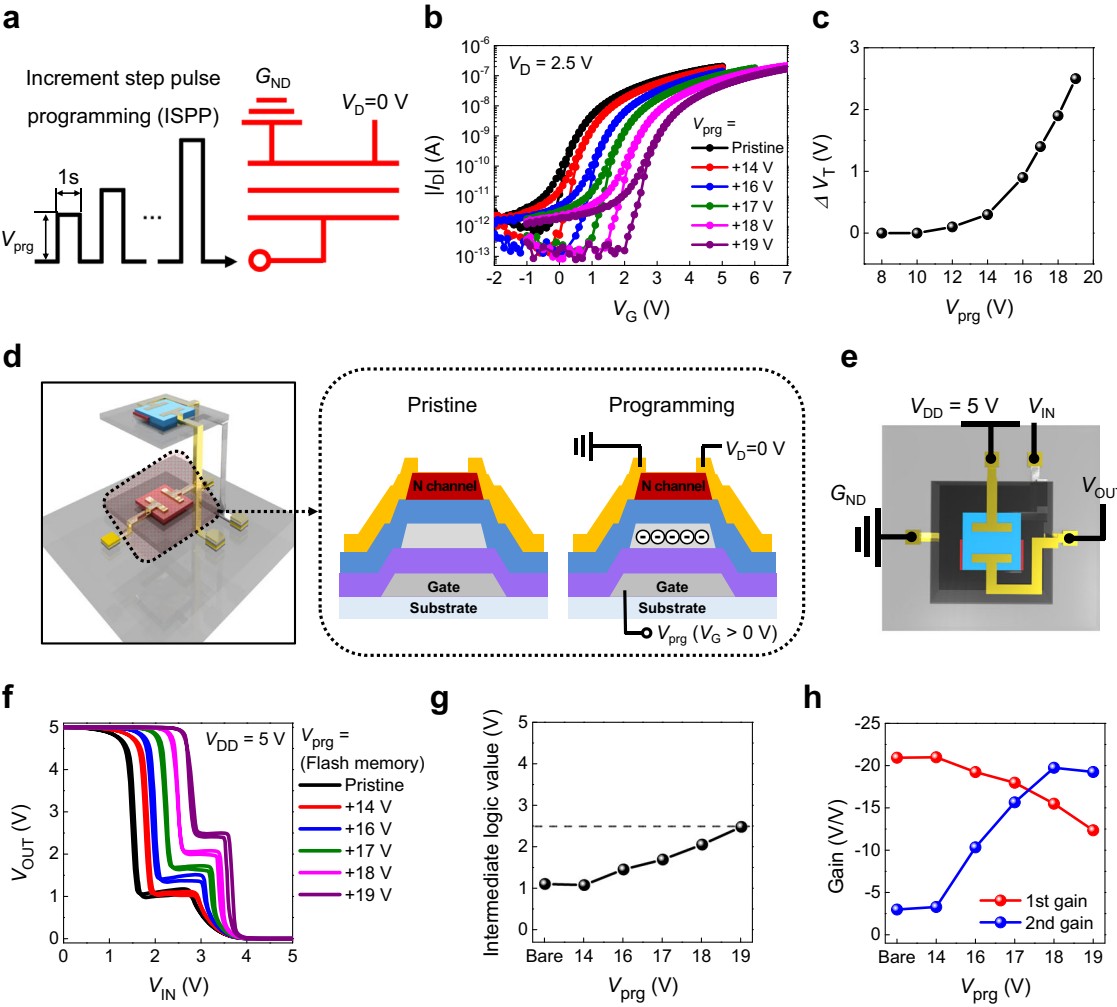

**Fig. 4 The 3-dimensional (3D) ternary logic inverter (T-inverter) performance according to the programming state of the flash memory. a** A schematic diagram of the incremental step pulse programming (ISPP) of the flash memory. **b** The transfer characteristics of the memory and **c** change in threshold voltage ($V_T$) with respect to a programming voltage ($V_{prg}$). A schematic illustration of **d** the 3D T-inverter with different flash memory states and **e** 3D T-inverter measurement. **f** The voltage transfer characteristics (VTCs), **g** intermediate logic value, and **h** DC gain values according to $V_{prg}$ of the flash memory.

value strongly suggests that the balance between channel conductance of the flash memory and HTR is important to achieving a high-performance T-inverter, which was achieved successfully in this study by replacing the n-type transistor with flash memory.

On the other hand, the transfer curve shifted gradually toward the negative direction in ISPE (Supplementary Fig. 19). The $\Delta V_T$ was −3.4 V in erasing operation, thus total memory window between the programming and erasing operations was calculated to be as large as 5.9 V. In contrast to the programming operation, the intermediate logic value decreased with the increasing $|V_{ers}|$ and even the full-swing operation could not be retained at higher $|V_{ers}|$, because the n-channel conductance of the flash memory overwhelmed the channel conductance of the HTR (Supplementary Figs. 19 and 20).

To verify the practical applicability of the proposed 3D T-inverter, we performed transient measurements with different memory states (Fig. 5a). The $V_{IN}$ pulse and $V_{OUT}$ response of the 3D T-inverter with the pristine and programmed flash memory is shown in Fig. 5b, c, respectively. Three different $V_{IN}$ values were applied, of which value was adjusted every 3 s. The 3D T-inverter with both states exhibited no notable hysteresis regardless of the $V_{IN}$ history throughout the measuring time. However, the

intermediate logic value was clearly enhanced in the programming state, compared to that observed in the pristine state. The switching speed of the T-inverter was also investigated (Supplementary Fig. 21 and Supplementary Table 2), which showed that the switching speed was comparable to those obtained frequently in the organic binary logic circuits[57–59]. It follows from the above observation that the electrical characteristics of the vertically stacked 3D T-inverter, particularly the intermediate logic state, could be controlled systematically by the programming/erasing states of the flash memory. The intermediate logic value became close to the optimum value ($V_{DD}/2$) with the proper programming state, without sacrificing other electrical characteristics such as full-swing operation and negligible hysteresis. The 3D T-inverter developed in this study can accommodate many important requirements for high-performance ternary logic circuits, including the optimized intermediate logic value, full-swing operation, high DC gain as well as low-voltage operation (Supplementary Table 1). The customizable channel conductance offered by the introduction of flash memory structure resulted in such optimized T-inverter performance. It is also worthwhile to mention that the T-inverter in the form of vertical stack was realized (Supplementary Table 1), which can further enhance the integration density. In addition, the electrical characteristics could

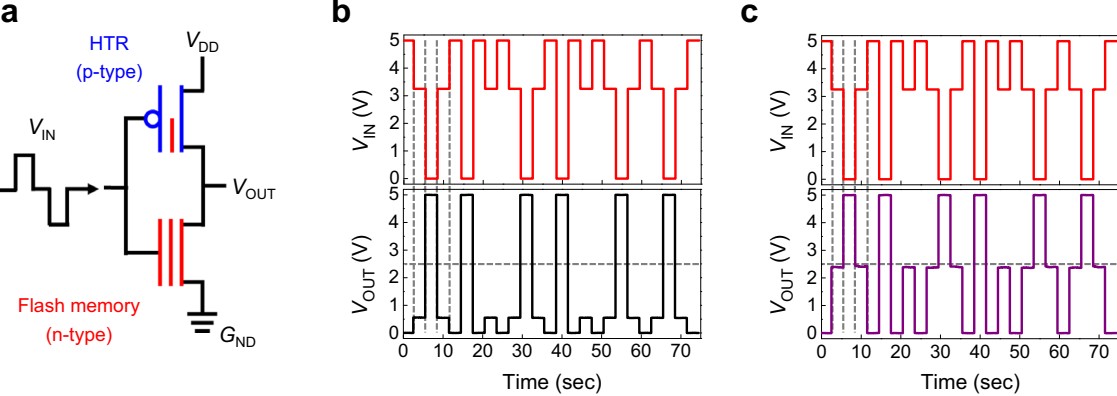

**Fig. 5 Transient measurement of the 3-dimensional (3D) ternary logic inverter (T-inverter). a** Schematic illustration of transient measurement of the 3D T-inverter and measurement results with **b** pristine and **c** programming states.

be adjusted with low $V_{prg}/V_{ers}$ as well as low operating voltage <5 V, resulting from high $\alpha_{CR}$ together with the ultrathin but high-performance polymer dielectric layers.

**Retention characteristics of the flash memory and T-inverter.** On top of the successful control and optimization of the intermediate logic state, the stability should also be secured to achieve a reliable long-term 3D T-inverter operation. In detail, it is imperative that the 3D T-inverter fully maintains its superior electrical characteristics over time, which is directly related to the retention characteristics of the flash memory. To investigate the memory retention, the transfer curves of the optimum programming state ($V_{prg} = +19$ V) were monitored with time (Fig. 6a). The flash memory showed negligible change in transfer characteristics over time with only the 0.7 V of $\Delta V_T$ even after $10^5$ s. The change in $I_D$ at $V_G = 1$ V and $V_D = 2.5$ V was extracted from the transfer curves with the programming/erasing states (Fig. 6b and Supplementary Fig. 22). The $I_D$ in on-state (erasing state) at $10^3$, $10^4$, and $10^5$ s showed 0.999, 0.911, and 0.716 times of the initial $I_D$, respectively, which indicates the $I_D$ change less than an order of magnitude even after $10^5$ s. Such long retention characteristics enabled the flash memory to retain sufficient $I_D$ difference in the programming/erasing states over time. To the best of our knowledge, the flash memory device developed in this study exhibited long retention characteristics as well as the lowest $V_{prg}/V_{ers}$ among the reported organic FG flash memories based on polymer dielectrics, resulting from the tunneling-based programming mechanism through the robust dielectric layers with high $\alpha_{CR}$ (Fig. 6c and Supplementary Table 3)[60–70]. Moreover, the flash memory showed the retention characteristics even superior or at least comparable to those obtained from the organic memories with different operating principles including charge trapping memories and photonic memories[71–78]. Accordingly, the 3D T-inverter fully maintained its three distinct logic states with full-swing operation over time owing to the long retention characteristic of the flash memory (Fig. 6d, e). Also, no notable device degradation was observed, such as hysteresis behavior. The intermediate logic values were kept close to the optimum value (2.45, 2.47, and 2.42 V at $10^3$, $3 \times 10^3$, and $10^4$ s, respectively) and became 2.18 V even after $10^5$ s (Fig. 6f). The intermediate logic value was slightly decreased with time, because the transfer curve of the flash memory was shifted to negative $V_G$ direction, which made $I_D$ of the flash memory higher than that of the HTR in the NTC region (Fig. 6d). Because of the long retention of three evidently discernable logic states, high DC gain values of the measured 3D T-inverter were maintained up to $10^5$ s (Supplementary Fig. 23). The 1st and 2nd gain values were also retained

to −13.6 and −20.8 V/V, respectively, and their change was <4.5 V/V for the 1st gain and 3.0 V/V for the 2nd gain over the whole range of the measuring time. The 3D T-inverter with erasing state of the flash memory also maintained its electrical characteristics, even though the full-swing operation could not be retained. The changes in the intermediate logic value and DC gain value <0.10 V and 5.5 V/V, respectively, even after $10^5$ s (Supplementary Fig. 22). It follows from the observations above that the electrical characteristics of the 3D T-inverter were fully preserved once the constituent flash memory was adequately programmed, mainly due to the long retention performance of the flash memory developed in this study.

The operational stability test of the 3D T-inverter with the memory programming state was also performed. The VTCs with 100 consecutive cycles are shown in Fig. 6g. The 3D T-inverter fully retained its initial VTC behavior, including full-swing operation with three discernable logic states even after 100 times of the consecutive operation. The $V_{OUT}$ values at $V_{IN} = 0$, 3.25, and 5 V were retrieved based on the number of cycles, corresponding to the logic "2", "1", and "0", respectively (Fig. 6h). Those values commonly showed negligible change throughout the entire cycles, which verifies the highly stable operation of the vertically stacked 3D T-inverter. The 3D T-inverter with the optimum programming state exhibited the exceptionally stable operation against time and repeated operation, even with the proposed vertically stacked structure.

## Discussion

A low-voltage organic ternary logic circuit, in which the organic HTR was vertically integrated with the organic nonvolatile flash memory, was demonstrated in this study. The integration of nonvolatile flash memory into T-inverter allowed for the circuit programmability, where the electrical characteristics including the intermediate logic state could be controlled precisely by the charge storage in FG. All the device fabrication processes were based on the solvent-free vapor-phase deposition methods, which enabled a high degree of uniformity in the device performance. By applying the via-hole-less, vertical interconnection scheme to the MVL circuit, the T-inverter in 3D structure was fabricated, which can enhance the device density per unit area. In the organic flash memory, the high-$k$ and low-$k$ polymer dielectric pairs were used as BDL and TDL, respectively, which could effectively reduce the $V_{prg}/V_{ers}$ as well as operating voltage (~5 V). The electrical characteristics of the 3D T-inverter were controlled systematically according to the programming/erasing state of the flash memory. With the optimum programming state of the flash memory, the 3D T-inverter showed three clearly distinguishable logic states

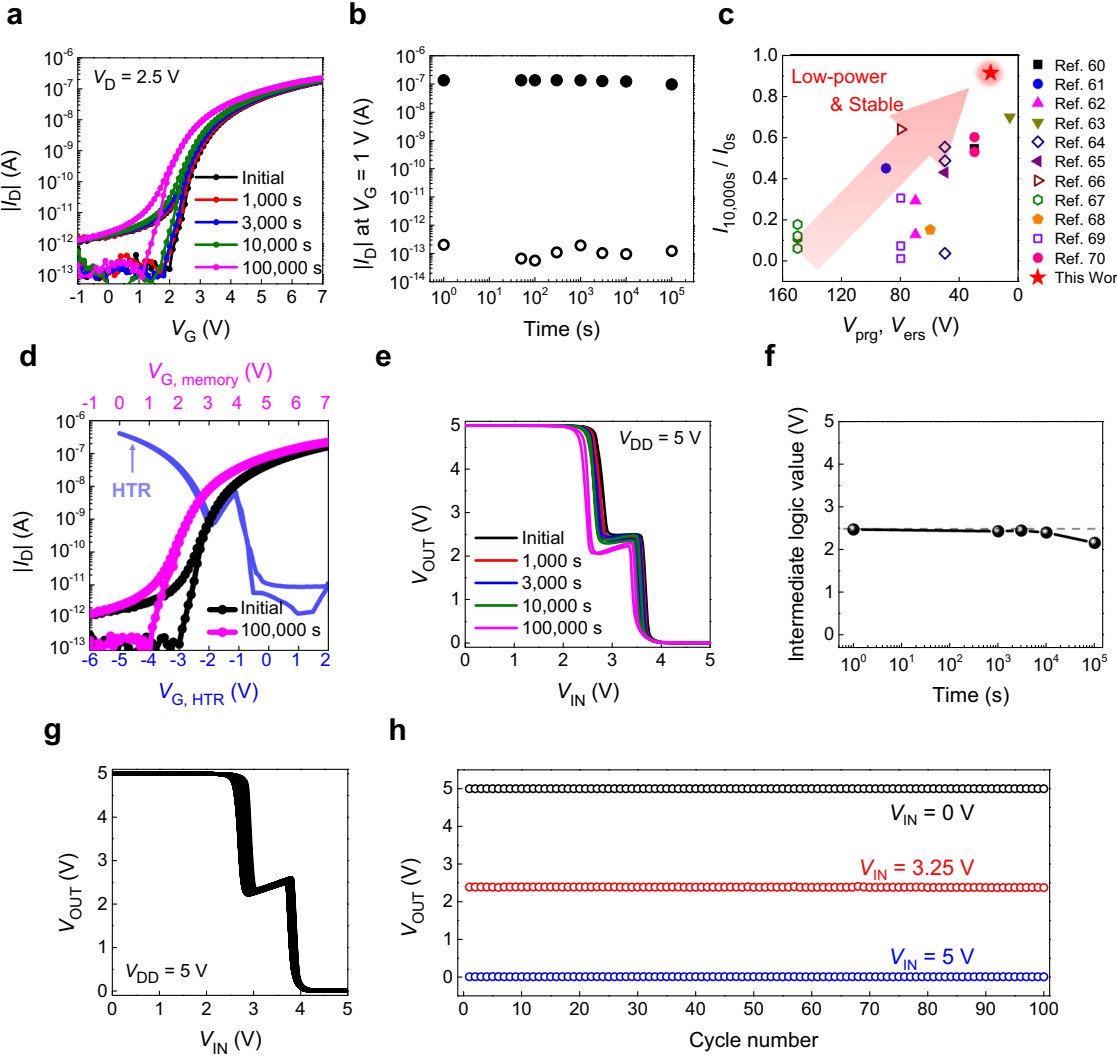

**Fig. 6 Stability of the three-dimensional (3D) ternary logic inverter (T-inverter). a** The change in transfer characteristics of the flash memory according to time after the optimum programming state (programming voltage ($V_{prg}$) = + 19 V) and **b** drain current ($I_D$) at gate voltage ($V_G$) = 1 V and $I_D$ = 2 V versus time in the programming (open symbol) and erasing (closed symbol) states. **c** The comparison of the retention characteristics and programming voltage/erasing voltage ($V_{prg}/V_{ers}$) of the flash memory in this study to those based on polymer dielectrics in the previous reports. **d** The change in transfer curves of the flash memory according to time and the transfer curve of the heterojunction transistor (HTR). **e** The change in voltage transfer characteristics (VTCs) and **f** intermediate logic value of the 3-dimensional (3D) ternary logic inverter (T-inverter) according to time. **g** The VTCs of 100 consecutive sweeps of the 3D T-inverter with the optimum programming states, and **h** output voltage ($V_{OUT}$) values at input voltage ($V_{IN}$) = 0, 3.25, and 5 V extracted from each cycle.

with the intermediate logic value close to the optimum one (~$V_{DD}$/2) as well as full-swing operation and low-voltage operation. The 3D T-inverter fully maintained its superior electrical characteristics and well-defined logic states over time, mainly resulting from the long retention characteristics of the flash memory. We believe the integration of the flash memory that enabled the controllable intermediate logic states in ternary logic circuits developed in this study can provide a useful insight to enhance the MVL circuit performance.

## Methods

**Materials and substrate**. For dielectric deposition, V3D3 (95%) and CEA ( >95%) monomers were purchased from Gelest, Tokyo Chemical Industry (TCI), respectively. DEGDVE (99%) and *tert*-butyl peroxide (TBPO) initiator (98%) were purchased from Sigma-Aldrich. For organic semiconductors, DNTT (99%) and PTCDI-C13 (98%) were purchased from Sigma-Aldrich. All the chemicals were used as received without further purification. The glass substrate of 25 mm × 25 mm (Samsung Corning Co.) was cleaned with ultrasonication by the subsequent

immersing in deionized (DI) water, acetone, and isopropyl alcohol for 20 min, followed by blowing with dry $N_2$ gas.

**Deposition of dielectric layers**. The dielectric layers were deposited by the custom-built iCVD system. For pV3D3 deposition, the V3D3 and TBPO were injected into the chamber at the flow rates of 2.5 and 1 standard cubic centimeter per minute (sccm)[40]. The chamber pressure and substrate temperature were maintained to 300 mTorr and 40 °C, respectively. For pC1D1 deposition, CEA, DEGDVE, and TBPO were injected with the flow rates of 0.28, 0.28 and 0.48 sccm, respectively[42]. The chamber pressure and substrate temperature were maintained to 60 mTorr and 30 °C, respectively. To decompose the initiator into the radicals, the filament was heated to 130 °C in the iCVD process. The V3D3, CEA, DEGDVE monomers were heated to 40, 50, and 50 °C, respectively, to vaporize. The dielectric layers were deposited through a shadow mask to achieve via-hole-less metal interconnection.

**Thin-film characterization**. To investigate the chemical composition of the dielectric layers, the XPS spectra were obtained by a Sigma probe multipurpose spectrometer (Thermo VG Scientific) with monochromatized Al Kα source. The AFM images were taken by using a scanning probe microscope (XE-100, Park Systems) to analyze the surface morphologies of the organic semiconductors and

polymer dielectric. The thickness of the polymer films and organic semiconductors was measured by a spectroscopic ellipsometer (M2000, Woollam).

**Device fabrication and characterization.** To fabricate the vertically stacked organic 3D T-inverter, the flash memory device was fabricated. For gate and FG electrode, Al was thermally evaporated with the deposition rate of 0.1 nm/s. The thickness of the gate and FG electrode was commonly set to 50 nm, which was monitored in situ by quartz crystal microbalance (QCM), and all the thermal evaporation processes were performed under a high vacuum lower than $2 \times 10^{-6}$ torr. The dielectric layers were deposited via iCVD process and the thicknesses of the pC1D1 BDL and pV3D3 TDL were 100 and 24 nm, respectively. The organic semiconductors including DNTT and PTCDI-C13 were also thermally deposited with a deposition rate of 0.03 nm/s. The PTCDI-C13 was recrystallized by the following thermal annealing at 200 °C for an hour. For $S/D$ electrodes, 70 nm-thick Au was thermally evaporated with a deposition rate of 0.1 nm/s. The channel width and length were 800 and 400 μm, respectively. 1 μm-thick pV3D3 was deposited onto the flash memory by the iCVD process as an ILD, and the HTR was fabricated directly on top of the ILD. For the fabrication of the HTR, 50-nm-thick Al electrode was thermally evaporated and used as gate electrode and the thickness of pV3D3 dielectric layer was 40 nm. The partial PTCDI-C13 semiconducting layer was deposited and recrystallized at 200 °C for one hour, followed by the DNTT layer that ranges from $S$ to $D$ electrode. 70 nm-thick Au $S/D$ electrode was deposited and the $D$ electrode of the HTR was connected to $D$ electrode of the flash memory through via-hole-less multi-metal interconnection. The overall areal dimension of the active area was 0.64 mm$^2$ (0.8 mm by 0.8 mm for width and length) including $S/D$ electrodes, and the areal dimension of the patterned dielectric layer was 1.44 mm$^2$ (1.2 mm by 1.2 mm for width and length). To visualize the 3D T-inverter, the device was sliced by a focused ion beam (Helios Nanolab 450), and cross-sectional HRTEM images were taken by Cs-corrected TEM (Titan cubed G2, FEI) with EDS mapping analysis. The insulating performance was analyzed by using metal-insulator-metal (MIM) devices where the polymer dielectric layers were deposited between thermally evaporated Al electrodes. The electrical characteristics were measured by B1500A semiconductor analyzer (Agilent Technologies). The sweeping speed of the input voltage was set to ~0.1 V/s, otherwise specified. The temperature of the devices was measured by infrared thermometer (Fluke 62 MAX). All the device fabrication and characterization were performed in the N$_2$-filled glovebox.

## Data availability

The data within the article and its supplementary information are available from the corresponding authors upon request.

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

## Acknowledgements

This work was supported by the Wearable Platform Materials Technology Center (WMC) funded by the National Research Foundation of Korea (NRF) Grant by the Korean Government (MSIT) (NRF-2022R1A5A6000846) (S.G.I.). This work was also supported by NRF grants funded by the MSIT, No. 2021R1A2B5B03001416 (S.G.I.) and No. 2020R1A2C1101647 (H.Y.).

## Author contributions

J.C., C.L., H.Y., and S.G.I. conceived the idea and designed the experiments. J.C. and C.L. designed, fabricated, and measured all the devices and circuits. C.L. assisted with device characterization and H.P. helped with the circuit design. S.M.L. assisted device fabrication and C.-H.K. performed thermal simulation analysis. J.C., C.L., H.Y., and S.G.I. wrote the manuscript. All authors reviewed the manuscript and discussed the results. J.C. and C.L. contributed equally to this work. H.Y. and S.G.I. contributed equally as corresponding authors.

## Competing interests

The authors declare no competing interests.

## Additional information

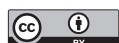

