## [Peer Review File · Nature Communications]

Reviewer comments, first round -

Reviewer #1 (Remarks to the Author):

This paper proposed an interesting solution to achieve multi-valued logic circuits via a combination of heterojunction transistor (HTR) and flash memory, and fabricated a 3-dimensional (3D) T-inverter with a vertically stacked form based on all-dry processes. This work is interesting, though the authors claimed outstanding device uniformity and stable performance were realized, the significance and the novelty of this work are not clear. For example, compared with ref 18 and other previous work, what are the technical advantages in the device fabrication? Further, what is the point to use a T-inverter for multi-valued logic circuits? The gain of such a device is generally lower compared with conventional binary inverters. The question is how to balance the number of logic values and the gain for each value? This paper now is not suitable for publication in Nature Communications, though a major revision is recommended. Additional points are:

1. On page 6, line 131: "a 1 μm -thick interlayer dielectric (ILD) was imposed between the unit devices to electrically isolate the flash memory and HTR.", will this interlayer hamper the heat dissipation of the underlying device and affect its stability? The author should track the temperature of the device during continuous operation.
2. Please carefully check the statement from line 137 to 140 on page 7.
3. On page 9, line 188 to 189, the author claimed that all on/off ratios are higher than 106, however, in figure 2c, it seems most of them are lower than 106, please double-check this part.
4. On page 10, line 206, please explain the BTBT in detail.
5. Please provide the output curves for both HTR and organic flash memory (at both pristine and programmed status, if possible). As these output curves would help to better understand the inverter ternary output transition curves at different states. Figure 1c in ref 18 is an ideal example for reference.
6. In figure 3f, the intermediate logic value window decreases notably with increasing V_{prg} , in other words, the intermediate step length at the "x" axis reduces with V_{prg} . What is the reason for this behavior? Please explain it in detail.
7. It is not convincing to use extrapolation to describe the retention time in Figure 4b. Moreover, the authors only show the I_{d} , intermediate logic values, and gain values at 10,000 s, which is relatively short. Please extend the retention time to 100,000 s (approximately 28 hours), which is doable in general (Adv. Mater, 2015, 27, 6257-6264, Nat. Mater, 2008, 7, 547-550.).
8. The author claimed an excellent retention characteristic on pages 14-15 based on the fact that the I_{d} maintained 0.911 times of the initial I_{d} at 10,000s. However, to our knowledge, this nearly 10% drop in I_{d} in less than 3 hours is a sign of moderate retention property (ACS Appl. Mater. Interfaces 2018, 10, 9563-9570; Nano Lett, 2009, 9, 1713-1719. Nat Comm, 2014, 5, 4720.). Further, the extrapolation method is not strictly scientific.

Reviewer #2 (Remarks to the Author):

In the manuscript, multi-valued logic inverter based on a heterojunction transistors and a nonvolatile floating-gate transistor were constructed with one heterojunction transistor stacked vertically on top of another nonvolatile floating-gate transistor. Intermediate V_{out} value of $\sim V_{\text{DD}}/2$, gain as high as 20 V/V were achieved, along with low driving voltage of less than 5 V. Moreover, long-term stability of the inverter was also obtained. This work is interesting and novel. It can be published in Nat. Comm. if the following comments can be properly addressed.

1. In the experimental section, details for the inverter fabrication are missing, such as the thicknesses of the floating gate electrode and other relevant layers, and layer dimensions. These parameters are extremely important for others to reproduce the work.
2. It is mentioned that a well-defined intermediate logic V_{out} state is important. How about the intermediate V_{in} value in T-inverters? Is the current work showing satisfactory intermediate V_{in} ?
3. The hysteresis of a transistor and inverter are highly dependent on the sweeping speed of the

input voltage. It is not accurate to say this transistor/inverter is hysteresis-free without mentioning the sweeping speeds.

4. Related to the last comments, how fast the inverter can be switched? What is the delay time? This is one important parameter for an inverter.

5. It is suggested to plot/draw the illustration of the inverter/transistor with both side view and top view, so that the structure of the devices can be much clearer.

Reviewer #3 (Remarks to the Author):

Excellent paper, very well structured, novel and interesting. The authors incorporate several novel ideas, including the use of new CVD-deposited insulators, and the use of these materials to create a flash memory. The paper further includes a comprehensive characterization of the fabricated devices, and uses more than enough devices and sweep measurements to confirm a scalable and reliable process. The paper also includes a reasonable background and cites the background work of other groups well.

I recommend for acceptance without any major modifications necessary.

Response to the Reviewers' Comments

The authors thank the Reviewers for their considerate review of our manuscript and the valuable comments. We have revised the manuscript and supporting information based on the Reviewers' comments. The Reviewers' comments appear in **black**, and the authors' responses in blue. In the revised manuscript, the changes with respect to the previous version are highlighted in yellow.

Reviewer (#1)'s COMMENTS:

This paper proposed an interesting solution to achieve multi-valued logic circuits via a combination of heterojunction transistor (HTR) and flash memory and fabricated a 3-dimensional (3D) T-inverter with a vertically stacked form based on all-dry processes. This work is interesting. Though the authors claimed outstanding device uniformity and stable performance were realized, the significance and the novelty of this work are not clear. For example, compared with ref 18 and other previous work, what are the technical advantages in the device fabrication? Further, what is the point to use a T-inverter for multi-valued logic circuits? The gain of such a device is generally lower compared with conventional binary inverters. The question is how to balance the number of logic values and the gain for each value? This paper now is not suitable for publication in *Nature Communications*, though a major revision is recommended.

Response:

We appreciate the Reviewer #1 for in-depth review of our manuscript and providing the valuable comments. We have carefully revised the manuscript by preparing the point-by-point response to each comment as follows.

General Comment #1) Though the authors claimed outstanding device uniformity and stable performance were realized, the significance and the novelty of this work are not clear. For example, compared with ref 18 and other previous works, what are the technical advantages in the device fabrication?

Response:

As the Reviewer #1 pointed out, our group previously demonstrated organic logic circuits by using via-hole-less metal interconnection to achieve high-density organic electronics (*IEEE Electron Device Lett.* **2020**, *41*, 1685-1687 (Ref. 18 in the manuscript) and *Nat. Commun.* **2019**, *10*, 2424). We would like to highlight the technical advances in the device fabrication achieved in this paper compared to our previous reports, as summarized in **Table R1**.

Table R1. Summary of the technical advances in organic circuits based on via-hole-less metal interconnection

	Nat. Commun. 2019.	IEEE Electron Device Lett. 2020.	This work
Unit device	Transistor (p, n)	Transistor (p, n)	Heterojunction transistor (p) Floating-gate flash memory (n)
Circuit configuration	Binary circuit	Binary circuit	Ternary logic circuit
Circuit programmability	X	X	O
Vertical integration	O	X	O
Uniformity (Number of devices)	20	12	22
Operating voltage	10 V	10 V	5 V

The primary technical advance in this report is the expansion of the applicability of the via-hole-less metal interconnection scheme into multi-valued logic circuits. Compared to the previous reports where conventional digital logic circuits were developed, we newly designed and demonstrated a ternary logic circuit in vertically stacked structure.

Especially in this study, an organic nonvolatile flash memory instead of a transistor was newly incorporated into the ternary logic inverter (T-inverter), which is, to the best of our knowledge, the first demonstration of the flash memory-integrated ternary logic circuit. The

incorporation of a nonvolatile flash memory allows for the systematic circuit programmability, where the precise control of the intermediate logic state in the T-inverter is enabled. With the new circuit design, the fabricated devices in this study clearly showed outstanding device performance, featured by well-defined three discernable logic states, full-swing operation as well as highly stable, hysteresis-free operation, which had not been achieved to date. Meanwhile, the ternary logic devices demonstrated in this report fully maintained the important advantages obtained in the previous binary logic circuits fabricated by using via-hole-less metal interconnection, such as the excellent device-to-device uniformity and low operating voltage (less than 5 V).

In addition, to achieve low programming/erasing voltage ($V_{\text{prg}}/V_{\text{ers}}$) of the flash memory as well as low operating voltage, we introduced a high- k dielectric layer, poly(2-cyanoethyl acrylate-*co*-diethylene glycol divinyl ether) (pCEA-*co*-DEGDVE) as a blocking dielectric layer (BDL) and verified its compatibility with the via-hole-less multi-metal interconnection architecture. Also, we employed an ultrathin tunneling dielectric layer (TDL) with via-hole-less metal interconnection, whose thickness was less than 30 nm, about a half of the dielectric thickness used in our previous reports.

Considering all these aspects, we believe the significance and novelty of this work can be justified.

We revised the manuscript as follows

Page 5, 2nd paragraph,

As an effective method to systematically control the electrical characteristics (i.e. channel conductance) of a transistor, a nonvolatile floating-gate (FG) flash memory was utilized and integrated with HTR to implement a T-inverter in this study. The implementation of the nonvolatile FG flash memory enabled a systematic circuit programmability, where the precise control of the threshold voltage (V_t) allows to enhance the T-inverter performance by optimizing the intermediate logic state. Furthermore, by utilizing the via-hole-less metal interconnection method,^{16, 18} the flash memory on the first floor was vertically integrated with the HTR on the second floor to realize a three dimensional (3D) T-inverter, which is, to the best of our knowledge, the first demonstration of a vertically stacked MVL circuit.

Page 6, 2nd paragraph,

In the 3D T-inverter fabrication process, a vapor-phase deposition method, termed initiated chemical vapor deposition (iCVD) process was introduced to deposit high-performance polymer dielectric layers while minimizing the damage to the underlying devices.⁴⁰⁻⁴³ The polymer dielectric materials were introduced with optimized dielectric layer configuration, which reduced the programming/erasing voltage ($V_{\text{prg}}/V_{\text{ers}}$), and enabled the operating voltage less than 5 V, which is comparable to or even lower than those obtained in the previous organic 3D logic circuits.¹⁴⁻¹⁶ Meanwhile, the excellent device-to-device uniformity was also fully retained. Most of all, the solvent-free deposition process was capable of in-situ patterning of the dielectric layer simply through shadow mask during the deposition process, which allowed etching-free, via-hole-less metal interconnection, and thus 3D vertical integration of the HTR and flash memory devices.¹⁶

Page 17, 3rd paragraph,

In summary, we demonstrated a low-voltage organic ternary logic circuit, in which the organic HTR was vertically integrated with the organic nonvolatile flash memory. The integration of a nonvolatile flash memory into T-inverter allowed for the circuit programmability, where the electrical characteristics including intermediate logic state can be controlled precisely by charge storage in FG. All the device fabrication processes were based on solvent-free vapor-phase deposition methods, which enables the high degree of uniformity in the device performance. By applying the via-hole-less, vertical interconnection scheme into the MVL circuit, the T-inverter in 3D structure was fabricated for the first time, which can enhance the device density per unit area. In the organic flash memory, the high- k and low- k polymer dielectric pairs were used as BDL and TDL, respectively, which could effectively reduce the $V_{\text{prg}}/V_{\text{ers}}$ as well as operating voltage (~ 5 V).

General Comment #2) Further, what is the point to use a T-inverter for multi-valued logic circuits? The gain of such a device is generally lower compared with conventional binary inverters. The question is how to balance the number of logic values and the gain for each value?

Response:

As Reviewer #1 commented, the gain value of the T-inverters can be inherently low compared to that of conventional binary inverters due to two different switching occurring in T-inverters. Nevertheless, ternary logic circuits based on HTRs exhibit a great advantage that the multi-logic circuits can drastically reduce the system complexity (*Nat. Nanotechnol.* **2017**, *12*, 1148-1154, *Nat. Mater.* **2017**, *16*, 170-181, *Nat. Commun.* **2019**, *10*, 1998 and *Adv. Sci.* **2021**, *8*, 2004216). Furthermore, please note that the DC gain is critically important in amplifier and high-speed buffer rather than arithmetic circuits as bandwidth in amplifier circuit and high-pass cutoff frequency are obtained by the degree of DC gain (*J. Low Power Electron. Appl.* **2019**, *9*, 26).

Moreover, compared to the previously reported ternary logic circuits, the gain values here can be regarded as one of the highest achieved to date, as summarized in **Table R2**. This information can also be found in **Supplementary Table 1**.

Table R2. Summary of the operating voltage and DC gain values in this study compared to the previous ternary logic circuits.

Reference	V_{IN}/V_{OUT} (V)	V_{DD} (V)	DC gain (V/V)
Nat. Commun. 2019 , 10 , No. 1998	5 / 5	5	>1
Nano Lett. 2018 , 18 , 4355-4359	4 / 8	10	N/A
Adv. Mater. 2019 , 31 , 1808265	50 / 50	50	>20
Nano Lett. 2016 , 16 , 1359-1366	1 / 0.9	1	4.5
Nat. Commun. 2016 , 7 , No. 13413	25 / 1.7	2	N/A
Nat. Nanotechnol. 2017 , 12 , 1148-1154	1 / 2	2	12
ACS Nano 2019 , 13 , 4478-4485	60 / 20	26	1.2
ACS Nano 2019 , 13 , 5430-5438	40 / 2	2	0.2

ACS Appl. Mater. & Interfaces 2020 , 12 , 36530-36539	80 / 8	8	N/A
Semicond. Sci. Technol. 2020 , 35 , 065020	80 / 10	10	1
Adv. Electron. Mater. 2020 , 6 , 2000426	2 / 2	2	N/A
This work	5 / 5	5	~20

Only one study reported the gain value comparable with this study (*Adv. Mater.* **2019**, *31*, 1808265). However, the operating voltage in that study was as high as 50 V, which is ten times higher than that achieved in this study, mainly due to the use of thick, rigid SiO₂ dielectric layer. In addition, we believe this paper can provide an important insight to optimize the T-inverter performance. For example, the 2nd gain value was dramatically enhanced from -3 to -19 V/V with the increased intermediate logic value by programming the n-type flash memory.

Even though ternary logic circuits are suitable to arithmetic circuit applications rather than amplifier or high-speed buffer as discussed above, the DC gain values obtained in this study (~20 V/V) are not quite low and comparable to many gain values achieved in the organic binary logic circuits with polymer dielectrics reported recently (*IEEE Trans. Electron Devices* **2014**, *61*, 1175-1180, *IEEE Trans. Electron Devices* **2014**, *61*, 2220-2223, *Adv. Electron. Mater.* **2017**, *3*, 1600557, *Org. Electron.* **2017**, *46*, 14-21, *Adv. Electron. Mater.* **2018**, *4*, 1700313, *Adv. Electron. Mater.* **2018**, *4*, 1800340, *Org. Electron.* **2019**, *75*, 105358, *J. Mater. Chem. C* **2020**, *8*, 15331-15338 and *ACS Appl. Mater. Interfaces* **2021**, *13*, 30921-30929) (Ref. 48-56 in the revised manuscript), where in most cases, a few tens of the DC gain values were obtained.

Considering the Reviewer #1's General Comment #2, we revised the manuscript by as follows:

Page 13, 1st paragraph,

Nevertheless, the obtained gain value was one of the highest DC gain values among the previously reported non-Si-based ternary logic inverters reported to date (Supplementary Table **1**).^{23, 29, 30, 32-34, 37} Also, the obtained gain values were

comparable to those observed in the organic binary logic inverters with polymer dielectric materials.⁴⁸⁻⁵⁶ Moreover, the dramatic increase of the 2nd gain value strongly suggests that the balance between channel conductance of the flash memory and HTR is important to achieve high-performance T-inverter, which was achieved successfully in this study by replacing the n-type transistor with a flash memory.

48. Feng L. et al. All-solution-processed low-voltage organic thin-film transistor inverter on plastic substrate. *IEEE Trans. Electron Devices* **61**, 1175-1180 (2014).

49. Feng L., Cui Q., Zhao J., Tang W. & Guo X. Dual-V_{th} low-voltage solution processed organic thin-film transistors with a thick polymer dielectric layer. *IEEE Trans. Electron Devices* **61**, 2220-2223 (2014).

50. Shiwaku R. et al. Printed organic inverter circuits with ultralow operating voltages. *Adv. Electron. Mater.* **3**, 1600557 (2017).

51. Shin E.-Y., Choi E.-Y. & Noh Y.-Y. Parylene based bilayer flexible gate dielectric layer for top-gated organic field-effect transistors. *Org. Electron.* **46**, 14-21 (2017).

52. Takeda Y. et al. Organic complementary inverter circuits fabricated with reverse offset printing. *Adv. Electron. Mater.* **4**, 1700313 (2018).

53. Stucchi E., Dell'Erba G., Colpani P., Kim Y. H. & Caironi M. Low-voltage, printed, all-polymer integrated circuits employing a low-leakage and high-yield polymer dielectric. *Adv. Electron. Mater.* **4**, 1800340 (2018).

54. Jo I. Y. et al. Low-voltage-operating complementary-like circuits using ambipolar organic-inorganic hybrid thin-film transistors with solid-state-electrolyte gate insulator. *Org. Electron.* **75**, 105358 (2019).

55. Stucchi E., Scaccabarozzi A. D., Viola F. A. & Caironi M. Ultraflexible all-organic complementary transistors and inverters based on printed polymers. *J. Mater. Chem. C* **8**, 15331-15338 (2020).

56. Park H. et al. Tailored polymer gate dielectric engineering to optimize flexible organic field-effect transistors and complementary integrated circuits. *ACS Appl. Mater. & Interfaces* **13**, 30921-30929 (2021).

Comment #1) On page 6, line 131: “a 1 μm -thick interlayer dielectric (ILD) was imposed between the unit devices to electrically isolate the flash memory and HTR.”, will this interlayer hamper the heat dissipation of the underlying device and affect its stability? The author should track the temperature of the device during continuous operation.

Response:

We appreciate the Reviewer #1 for the important comments. To clarify the heat dissipation of the underlying floating-gate device, we carried out a two-dimensional (2D) finite-element simulation by reproducing the fabricated devices including 1 μm -thick polymeric interlayer dielectric (ILD). The simulation result revealed that the temperature rise was within a level of 0.1 K, despite the low thermal conductivity of the organic materials employed for the device and the use of a relatively thick ILD polymer on the flash memory device. Such an observation is mainly due to the low operating voltage (~ 5 V).

Moreover, as Reviewer #1 suggested, thermal analysis was also performed experimentally by stacking 1 μm -thick pV3D3 ILD on top of the fabricated PTCDI-C13 flash memory. We measured the temperature using infrared (IR) thermometer under the continuous operation at $V_G = V_D = 5$ V and only the negligible temperature variation was observed throughout the continuous operation up to 5 hours.

We revised the manuscript by adding thermal analysis results as below:

Supplementary Figure 5 | Simulated physical quantities inside a PTCDI-C13 transistor at $V_G=V_D=5$ V. **a**, electron concentration, **b**, Joule heat power, and **c**, lattice temperature. **d**, Dependence of the maximum lattice temperature on the driving voltages.

Supplementary Figure 6 | **a**, Photography for the measurement of the device temperature with an hour interval. **b**, The change in I_D and temperature of PTCDI-C13 flash memory with 1 μ m-thick ILD during continuous operation at $V_G=V_D=5$ V.

A discussion on this aspect was added accordingly:

Page 9, 1st paragraph,

Furthermore, the average threshold voltage (V_T) was less than 0.6 V with narrow distribution (0.55 ± 0.09 V), thus ensuring the uniform channel conductance of the pull-down transistor. **It is also worthwhile to note that even with a relatively thick polymer ILD with low thermal conductivity, only negligible change in the temperature**

of the flash memory was observed during the continuous operation at 5 V, owing to the low operating voltage (Supplementary Fig. 5 and 6).

Details of the temperature measurement was added in Method section:

Page 28, in **Device fabrication & characterization** section,

The electrical characteristics were measured by B1500A semiconductor analyzer (Agilent Technologies). The sweeping speed of the input voltage was set to ~ 0.1 V/s, otherwise specified. The temperature of the devices was measured by infrared (IR) thermometer (Fluke 62 MAX). All the device fabrication and characterization were performed in the N_2 -filled glovebox.

More detailed discussion for thermal analysis result and the related references were added in Supplementary Information:

Page 7 in Supplementary Information,

Thermal analysis of the flash memory under polymer ILD

To clarify the heat dissipation of the underlying floating-gate device, we performed a two-dimensional finite-element simulation to investigate the thermal properties of the proposed system.⁵ The thermal boundary condition was set to 300 K at metal electrodes and the thermal conductivity and heat capacity of all organic materials (PTCDI-C13, pV3D3, pC1D1) was assumed to be 10^{-3} $Wcm^{-1}K^{-1}$ and 1 $Jcm^{-3}K^{-1}$, respectively.⁶ Supplementary Fig. 5a-c present the simulated physical quantities inside the PTCDI-C13 thin film at $V_G = V_D = 5$ V. A strong accumulation of electrons at the PTCDI-C13/pV3D3(TDL) interface is clearly identifiable in Supplementary Fig. 5a with an apparent channel pinch-off at the drain. This in turn resulted in pronounced Joule heating over the pinch-off region due to an elevated electric field, as shown in Supplementary Fig. 5b. Supplementary Fig. 5c shows the internal temperature distribution, where the direction and degree of heat dissipation from the thermal hot spot is distinctively visualized. Most importantly, despite the low thermal conductivity of organic materials and the use of a relatively thick interlayer dielectric polymer film

on top of the device, the temperature rise was practically negligible – no larger than 0.1 K with the low operating voltage (5 V), as illustrated in Supplementary Fig. **5d**.

Moreover, the temperature of the PTCDI-C13 flash memory fabricated with 1 μ m-thick pV3D3 ILD was measured under the continuous operation at $V_G = V_D = 5$ V. As shown in Supplementary Fig. **6**, the temperature change less than 0.1 K was observed throughout the whole continuous operation up to 5 hours, which is fully consistent with the result obtained by the two-dimensional finite-element simulation.

5. Han H. & Kim C.-H. Unexpected Benefits of Contact Resistance in 3D Organic Complementary Inverters. *Adv. Electron. Mater.* **6**, 1900879 (2020).

6. Wang, H. & Yu, C. Organic Thermoelectrics: Materials Preparation, Performance Optimization, and Device Integration. *Joule* **3**, 53-80 (2019).

Comment #2) Please carefully check the statement from line 137 to 140 on page 7.

Response:

We appreciate Reviewer #1's attentive review. We also rechecked the whole manuscript thoroughly and corrected all errors and typos as much as we can (including Response to Comment #3).

We revised the manuscript as follows:

Page 7, 1st paragraph,

Note that the drain electrode on top of PTCDI-C13/DNTT heterojunction in HTR was interconnected to the drain electrode in the flash memory and used as an output electrode in the inverter operation to induce an intermediate logic state.²⁹

Page 27, in **Device fabrication & characterization** section,

Device fabrication & characterization

For *S/D* electrodes, 70 nm-thick Au was thermally evaporated with the deposition rate of 0.1 nm/s. The channel width and length were 800 and 400 μm , respectively. 1 μm -thick pV3D3 was deposited onto the flash memory as an ILD and the HTR was fabricated directly on top of the ILD.

The scale bars in the optical microscope (OM) images were corrected accordingly:

Page 9, in Supplementary Information,

Supplementary Figure 7 | The OM images (left) and AFM images (right) at the middle of the active layer regions of **a**, the flash memory, **b**, ILD deposited on top of the flash memory and **c**, HTR.

Comment #3) On page 9, line 188 to 189, the author claimed that all on/off ratios are higher than 10^6 , however, in figure 2c, it seems most of them are lower than 10^6 , please double-check this part.

Response:

We appreciate Reviewer #1's critical review comments. We rechecked the raw data and found that the on/off ratio was slightly lower than 10^6 . We revised the manuscript accordingly. The on/off current value at each current level and the calculated on/off current ratio (I_{on}/I_{off}) therefrom are also summarized in **Table R4**.

Table R4. The I_{off} , I_{on} and I_{on}/I_{off} values of the organic flash memory

Device #	1	2	3	4	5	6	7	8	9	10	11
I_{off} ($\times 10^{-13}$ A)	2.63	4.47	3.37	3.08	3.39	3.60	2.25	3.24	4.20	3.08	2.30
I_{on} ($\times 10^{-7}$ A)	1.97	2.02	1.99	2.01	1.97	1.90	1.93	1.90	1.97	1.90	1.98
I_{on}/I_{off} ($\times 10^5$)	7.50	4.51	5.89	6.51	5.80	5.27	8.58	5.87	4.70	6.17	8.63

We revised the manuscript as follows:

Page 9, 1st paragraph,

As shown in the statistical analysis, all the flash memory devices exhibited on/off current ratio (I_{on}/I_{off}) higher than 10^5 with the operating voltage less than 5 V (Fig. **2c**).

Comment #4) On page 10, line 206, please explain the BTBT in detail.

Response:

We appreciate Reviewer #1 for the constructive suggestion. To elucidate the BTBT in more detail, we added a discussion on the operating principle of the HTR by dividing the operation states in the transfer curve into three regimes.

We revised the manuscript as follows:

Page 10, 1st paragraph,

All the HTR exhibited NTC characteristics with four different operation regions: (i) off-state ($-0.8 \text{ V} < V_G < 2 \text{ V}$), (ii) subthreshold region ($-1.5 \text{ V} (= V_{\text{peak}}, \text{ peak voltage}) < V_G < -0.8 \text{ V}$) where both hole-induced current and electron-induced band-to-band tunneling (BTBT) contributed to drain current (I_D). (iii) NTC region ($-2.6 \text{ V} (= V_{\text{valley}}, \text{ valley voltage}) < V_G < -1.5 \text{ V}$) where a depletion of n-type PTCDI-C13 occurred, thus $|I_D|$ was decreased with the increasing $|V_G|$, and (iv) on-state ($-5 \text{ V} < V_G < -2.6 \text{ V}$) with more hole accumulation in the higher $|V_G|$ range (Fig. 2e). A more detailed description for the origin of I_D in each regime of HTR can be found in Supplementary Fig. 8. All the HTR devices exhibited distinct NTC behaviors in the transfer characteristic.

We added detailed description of the operating principle of the HTR:

Supplementary Figure 8 | The operating principle of the HTR divided by three distinguishable regimes.

A discussion was added accordingly:

Page 10 in Supplementary Information,

The operating principle of the HTR

The HTR has a structure in which one electrode forms a contact with a stacked n-type/p-type semiconductor and the other electrode contacts a p-type semiconductor only. In the p-n junction at the channel, the band-to-band tunneling (BTBT) occurs as

the conduction band in the intrinsic region aligns with the valence band in the p-type region.^{8, 9} In the valence band of the p-type region, electrons tunnel into the conduction band of the intrinsic region and current can flow across the device.

The observed negative differential transconductance (NTC) implies the existence of the BTBT current. The operation principle can be divided into three different regions, as shown in Supplementary Fig. 8:

i) $0 \text{ V} > V_G > -1.5 \text{ V}$: Electron BTBT current occurs, and hole carriers begin to accumulate. Both the BTBT current and p-channel current contribute to I_D , exhibiting a peak current ($3.5 \times 10^{-8} \text{ A}$ at $V_G = -1.5 \text{ V}$).

ii) $-1.5 \text{ V} > V_G > -2.8 \text{ V}$: The BTBT current decreased as the n-type PTCDI-C13 is depleted by the negative gate voltage bias, resulting in the decrease of I_D with a valley current ($2 \times 10^{-9} \text{ A}$ at $V_G = -2.8 \text{ V}$).

iii) $V_G < -2.8 \text{ V}$: The n-type PTCDI-C13 is completely depleted, and only the p-channel current contributes to I_D .

The references regarding BTBT were added:

8. Yoo H., On S., Lee S. B., Cho K. & Kim J. J. Negative Transconductance Heterojunction Organic Transistors and their Application to Full-Swing Ternary Circuits. *Adv. Mater.* **31**, 1808265 (2019).

9. Nourbakhsh A., Zubair A., Dresselhaus M. S. & Palacios T. Transport properties of a MoS₂/WSe₂ heterojunction transistor and its potential for application. *Nano Lett.* **16**, 1359-1366 (2016).

Comment #5) Please provide the output curves for both HTR and organic flash memory (at both pristine and programmed status, if possible). As these output curves would help to better understand the inverter ternary output transition curves at different states. Figure 1c in ref 18 is an ideal example for reference.

Response:

We appreciate Reviewer #1 for the constructive suggestion. As Reviewer #1 suggested, we added the output curves for both HTR and organic flash memory. For the flash memory, output curves for both pristine and programmed states are added, which supports that the I_D of the flash memory was successfully modulated according to the memory programming states.

We revised the manuscript as follows:

Page 12, 2nd paragraph,

The electrical characteristics of the vertically stacked 3D T-inverter were analyzed according to the programming/erasing states of the flash memory on the 1st floor (Fig. **3d** and **e**). The I_D of the flash memory was modulated successfully by programming operation (Supplementary Fig. **15**), and with the decreasing channel conductance by charge trapping into the FG, the intermediate logic state was systematically controlled as shown in the VTC of the 3D T-inverter (Fig. **3f**). The intermediate logic value was gradually increased in accordance with the increasing V_{prg} and reached to 2.49 V with $V_{prg}=+19$ V (Fig. **3g**).

We added the output curves of the HTR and organic flash memory:

Supplementary Figure 15 | The output curves of the HTR (blue) and flash memory (red) in **a**, pristine and **b**, the programmed state ($V_{prg}=+19$ V).

Discussion was added accordingly:

Page 17 in Supplementary Information,

Output characteristics of the flash memory and HTR

The output characteristics of the HTR and flash memory with each programming state is shown in Supplementary Fig. **15**. In the flash memory, I_D decreased significantly after applying V_{prg} of +19 V compared to that observed in the pristine flash memory, which supports that the channel conductance was controlled successfully by the programming operation of the flash memory.

Comment #6) In figure 3f, the intermediate logic value window decreases notably with increasing V_{prg} , in other words, the intermediate step length at the “x” axis reduces with V_{prg} . What is the reason for this behavior? Please explain it in detail.

Response:

We appreciate Reviewer #1’s valuable comment. To investigate the intermediate logic state in the T-inverter, we analyzed the current overlap between HTR and flash memory (Supplementary Fig. **17**). At $V_{\text{prg}} = +14, +16$ V, the V_G regions where the flash memory and HTR showed similar channel conductance (I_D difference less than an order of magnitude) and corresponding V_{IN} range of the intermediate logic state was relatively wide (Supplementary Fig. **17a** and **b**). Nevertheless, the I_D of the flash memory is slightly higher than that in the NTC region of HTR in this range, which caused the V_{OUT} value (1.09 and 1.47 V at $V_{\text{prg}} = +14$ and +16 V, respectively) a bit lower than ideal value ($V_{\text{DD}}/2 \sim 2.5$ V) (Supplementary Fig. **17e** and **f**). On the other hand, at the increased V_{prg} of +19 V, I_D of the flash memory became practically identical to that of HTR in the NTC region, which resulted in the intermediate logic value of 2.49 V that is close to the ideal V_{OUT} value ($V_{\text{DD}}/2 \sim 2.5$ V, Supplementary Fig. **17d** and **h**). Therefore, the V_{IN} range exhibiting the ideal intermediate logic value in T-inverters is eventually determined by the NTC region of the HTRs, and we presented a strategy in this study that can utilize the NTC region as an ideal intermediate logic value.

We revised the manuscript by adding the detailed analysis of the transistor characteristics of the flash memory with different V_{prg} and the HTR with different V_{D} :

Supplementary Figure 17 | The overlapped transfer characteristics of the flash memory according to V_{prg} and the HTR with different V_{D} ; **a**, $V_{\text{prg}}=+14$ V, $V_{\text{D,HTR}}=-4$ V, **b**, $V_{\text{prg}}=+16$ V, $V_{\text{D,HTR}}=-3.5$ V, **c**, $V_{\text{prg}}=+18$ V, $V_{\text{D,HTR}}=-3$ V and **d**, $V_{\text{prg}}=+19$ V, $V_{\text{D,HTR}}=-2.5$ V. **e-h**, The corresponding VTC of the T-inverter.

A discussion was added accordingly:

Page 13, 1st paragraph,

The intermediate logic value was gradually increased in accordance with the increasing V_{prg} and reached to 2.49 V with $V_{\text{prg}}=+19$ V (Fig. **3g**). Therefore, the optimum programming state of the flash memory allowed that the intermediate logic value is practically identical to $V_{\text{DD}}/2$ (logic "1"), which made the intermediate logic state clearly distinguishable from V_{DD} (logic "2") and G_{ND} (logic "0"). The V_{IN} range where the intermediate logic states appeared was gradually shifted with the increasing V_{prg} , because the NTC region of the HTR was shifted to lower $|V_{\text{G}}|$ region with the decreasing $|V_{\text{D}}|$ (Supplementary Fig. **16**). The V_{IN} range of the intermediate

logic states decreased slightly with the increasing V_{prg} , because only the NTC region in the HTR could be represented to the intermediate logic state, which enabled us to obtain the ideal V_{out} value ($V_{\text{DD}}/2$) by optimizing the memory programming process (Supplementary Fig. 17). The 1st gain decreased gradually while 2nd gain increased with the increasing V_{prg} , resulting from the increased intermediate logic value (Fig. 3h and Supplementary Fig. 18).

Page 19 in Supplementary Information,

The overlapped transfer curves of the transistors and resulting VTC of the T-inverter according to memory programming state

Supplementary Fig. 17a-d shows the overlapped transfer curves of the flash memory and HTR to analyze the intermediate logic state according to the memory programming state. The VTCs of the T-inverter was also shown in Supplementary Fig. 17e-h along with each memory programming state. With the low V_{prg} (+14 and +16 V), the V_{G} region where the flash memory and HTR exhibited similar I_{D} (less than an order of magnitude difference) was relatively wide (~ 1.2 V) (Supplementary Fig. 17a, b). In those V_{G} regions, the flash memory showed slightly higher I_{D} compared to that of the HTR, resulting in low intermediate logic value (1.09 and 1.47 V at $V_{\text{prg}} = +14$ and +16 V, respectively). However, with the V_{prg} of +19 V, both transistors showed quite similar I_{D} values in the NTC of the HTR, which produced the ideal V_{OUT} value ($\sim V_{\text{DD}}/2$) in the intermediate logic state. In this optimum programming state, the V_{IN} range of the intermediate logic state decreased slightly to ~ 0.9 V, because only the NTC region in the HTR can represent the intermediate logic state.

Comment #7) It is not convincing to use extrapolation to describe the retention time in Figure 4b. Moreover, the authors only show the I_{a} , intermediate logic values, and gain values at 10,000 s, which is relatively short. Please extend the retention time to 100,000 s (approximately 28 hours), which is doable in general (*Adv. Mater.* **2015**, 27, 6257-6264, *Nat. Mater.* **2008**, 7, 547-550.).

Response:

We appreciate Reviewer #1's constructive comment. As Reviewer #1 recommended, we measured the electrical characteristics of the devices with the extended retention time up to 100,000 s. However, the $I_{10,000s}/I_{0s}$ values on the y-axis in Fig. **4c** was used instead of $I_{100,000s}/I_{0s}$, to accommodate the previous studies as many as possible for comparison of the retention performance, because $I_{10,000s}$ rather than $I_{100,000s}$ information was presented in most of the references. Instead, we added a summary of the retention characteristics of the flash memory developed in this study compared to previously reported organic floating-gate flash memories adapting polymer dielectrics, by using both $I_{10,000s}/I_{0s}$ and $I_{100,000s}/I_{0s}$ (Supplementary Table 3).

We fully agree with Reviewer #1 that the long-term retention by extrapolation is not an experimental value but a predicted one, even though we tried our best to extract as strictly as possible. Therefore, we deleted long-term retention by extrapolation, and revised x-axis in Figure 4b so that the measured value could be seen clearly.

We revised the manuscript by adding the change in the electrical characteristics of the devices with the prolonged time:

Figure 4 | Stability of the 3D T-inverter. **a**, The change in transfer characteristics of the flash memory according to time after the optimum programming state ($V_{\text{prg}} = +19$ V) and **b**, I_D at $V_G = 1$ V and $I_D = 2$ V versus time in the programming (open symbol) and erasing (closed symbol) states. **c**, The comparison of the retention characteristics and $V_{\text{prg}}/V_{\text{ers}}$ of the flash memory in this study to those based on polymer dielectrics in the previous reports. **d**, The change in transfer curves of the flash memory according to time and the transfer curve of the HTR. **e**, The change in VTCs and **f**, intermediate logic value of the 3D T-inverter according to time.

Supplementary Figure 22 | **a**, The change in transfer curves of the flash memory, **b**, VTCs and **c**, DC gain profile of the 3D T-inverter according to time. **d**, The intermediate logic value, **e**, 1st and 2nd gain values versus time extracted from VTCs.

Supplementary Figure 23 | The DC gain profiles of the 3D T-inverter with the optimum memory programming state ($V_{prg}=+19$ V) according to time.

We revised the manuscript accordingly:
 Page 15, 2nd paragraph,

To investigate the memory retention, the transfer curves of the optimum programming state ($V_{\text{prg}}=+19$ V) were monitored with time (Fig. **4a**). The flash memory showed negligible change in transfer characteristics over time with only the **0.7 V** of ΔV_T even after **10^5 s**. The change in I_D at $V_G=1$ V and $V_D=2.5$ V was extracted from the transfer curves with the programming/erasing states (Fig. **4b** and Supplementary Fig. **22**). The I_D in on-state (erasing state) at **10^3 , 10^4 and 10^5 s** showed **0.999, 0.911 and 0.716 times** of the initial I_D , respectively, **which indicates the I_D change less than an order of magnitude even after 10^5 s**. Such excellent retention characteristics enabled the flash memory to retain sufficient I_D difference in the programming/erasing states over time. To the best of our knowledge, the flash memory device developed in this study exhibited one of the best retention characteristics as well as lowest $V_{\text{prg}}/V_{\text{ers}}$ among the organic FG flash memory based on polymer dielectrics, resulting from the tunneling-based programming mechanism through the robust dielectric layers with high α_{CR} (Fig. **4c** and **Supplementary Table 3**).⁶⁰⁻⁷⁰

Page 16, 1st paragraph,

Accordingly, the 3D T-inverter fully maintained its three distinct logic states with full-swing operation over time owing to the excellent retention performance of the flash memory (Fig. **4d** and **4e**). Also, no notable device degradation was observed, such as hysteresis behavior. **The intermediate logic values were kept close to the ideal value (2.45, 2.47 and 2.42 V at 10^3 , 3×10^3 and 10^4 s, respectively) and became 2.18 V even after 10^5 s (Fig. **4f**).** The intermediate logic value was slightly decreased with time, **because the transfer curve of the flash memory was shifted to negative V_G direction, which made I_D of the flash memory higher than that of the HTR in the NTC region (Fig. **4d**).** Because of the excellent retention of three evidently discernable logic states, **high DC gain values of the measured 3D T-inverter were maintained up to 10^5 s (Supplementary Fig. **23**).** The 1st and 2nd gain values were also retained to **-13.6 and -20.8 V/V, respectively, and their change was less than 4.5 V/V for the 1st gain and 3.0 V/V for the second gain over the whole range of the measuring time.** The 3D T-

inverter with erasing state of the flash memory also maintained its electrical characteristics, even though the full-swing operation could not be retained. The changes in the intermediate logic value and DC gain value less than 0.10 V and 5.5 V/V, respectively, even after 10^5 s (Supplementary Fig. 22). It follows from the observations above that the electrical characteristics of the 3D T-inverter were fully preserved once the constituent flash memory was adequately programmed, mainly due to the excellent retention performance of the flash memory developed in this study.

Page 27 in Supplementary Information,

The retention characteristics of the flash memory in erasing operation

The retention characteristics in memory erasing operation were also investigated. The change in transfer characteristics with time after applying the maximum V_{ers} (-8 V) are shown in Supplementary Fig. 22a, which showed negligible change over time with only 0.6 V of ΔV_T after 10^5 s. This excellent retention performance led little change in the electrical characteristics of the 3D T-inverter over time (Supplementary Fig. 22b-c). Also, there was no notable device degradation such as hysteresis behavior. The change in intermediate logic value was less than 0.10 V even after 10^5 s (Supplementary Fig. 22d). Even with the extremely small change in the transfer curves, V_{OUT} at $V_{\text{IN}}=0$ V was recovered from 4.45 to 4.78 V in the VTC after 10^5 s, which induced the enhanced voltage swing and the improved 1st gain value from -17.7 to -23.1 V/V (Supplementary Fig. 22e). Nevertheless, there was still negligible change in the 2nd gain value (less than 0.5 V/V) throughout the whole measurement time.

Page 29 in Supplementary Information,

The DC gain profiles of the flash memory in programming state over time

The changes in DC gain profiles of the flash memory with the optimum programming state ($V_{\text{prg}}=+19$ V) are shown in Supplementary Fig. 23. Only negligible change was

observed in the DC gain values (less than 4.5 and 3.0 V/V for the 1st and 2nd gain values, respectively) and their V_{IN} positions (less than 0.35 V) even after 10^5 s.

We added the table that summarizes the retention characteristics of the organic floating-gate flash memory:

Page 28 in Supplementary Information,

Comparison of the retention characteristics

Supplementary Table 3 summarizes the retention characteristics of the flash memory developed in this study compared to the previously reported organic flash memories employing polymer dielectric materials.

Supplementary Table 3 | Summary of the retention characteristics of the reported floating-gate flash memory based on polymer dielectric.

Year ^[reference]	V_{prg}, V_{prg} (V)	$I_{10,000s}/I_{0s}$	$I_{100,000s}/I_{0s}$
2009 ^[60]	30	0.544	N/A
2010 ^[61]	90	0.448	0.316
2013 ^[62]	70	0.291	N/A
	70	0.126	
2014 ^[63]	6	0.697	0.612
	50	0.552	
2015 ^[64]	50	0.485	N/A
	50	0.035	
2015 ^[65]	50	0.428	N/A
2015 ^[66]	80	0.637	0.386
	150	0.175	
2016 ^[67]	150	0.119	N/A
	150	0.109	
	150	0.059	
2018 ^[68]	60	0.149	N/A
2018 ^[69]	80	0.303	N/A
	80	0.071	

	80	0.010	
2021 ^[70]	30	0.599	N/A
	30	0.528	
This work	19	0.911	0.716

Comment #8) The author claimed an excellent retention characteristic on pages 14-15 based on the fact that the I_d maintained 0.911 times of the initial I_d at 10,000s. However, to our knowledge, this nearly 10% drop in I_d in less than 3 hours is a sign of moderate retention property (*ACS Appl. Mater. Interfaces* **2018**, *10*, 9563-9570; *Nano Lett.*, **2009**, *9*, 1713-1719. *Nat Comm.*, **2014**, *5*, 4720.). Further, the extrapolation method is not strictly scientific.

Response:

We appreciate Reviewer #1's constructive comment. However, we still believe that the organic flash memory developed in this study showed excellent retention performance. As shown in **Table R5**, the I_D change was less than an order of magnitude throughout measuring time, which is clearly one of the highest retention performances compared to the previously developed organic floating-gate flash memories employing polymer dielectric materials (Fig. **4c** and Supplementary Table **3**).

Table R5. The I_D values measured at $V_G=1$ V according to time.

Time (s)	0	50	100	300	1,000	3,000	10,000	100,000
I_D at $V_G=1$ V ($\times 10^{-7}$ A)	1.347	1.351	1.352	1.355	1.345	1.264	1.228	0.965

We also carefully checked the retention performance in the previous reports that Reviewer #1 mentioned. Also, we compared our retention performance to other organic memories based on charge trapping mechanism, which are summarized in **Table R6**.

Table R6. Summary of the operating voltage and DC gain values in this study compared to the previous ternary logic circuits

Reference	Memory operating principle	V_{prg} or V_{ers} (V)	$I_{10,000\text{s}}/I_{0\text{s}}$	$I_{100,000\text{s}}/I_{0\text{s}}$
Nat. Mater. 2008 , 7, 547-550	Ferroelectric	20	0.768	0.632
ACS Appl. Mater. Interfaces 2018 , 10, 9563-9570	Electrical double-layer	9	0.334	0.492
Nat Commun. 2014 , 5, 4720.	Photonic flash memory	40	0.605	0.249
Adv. Mater. 2015 , 27, 6257-6264	Charge trapping	50	0.912	0.639
Adv. Funct. Mater. 2020 , 30, 2004665	Charge trapping	16	0.698	0.576
ACS Appl. Mater. Interfaces 2015 , 7, 10957-10965	Charge trapping	30	0.454	N/A
Chem. Asian. J. 2016 , 11, 1631-1640	Charge trapping	40	0.394	N/A
Adv. Electron. Mater. 2019 , 5, 1800799	Charge trapping	14	0.340	0.078
This work	Flash memory	19	0.911	0.716

As summarized in **Table R6**, our flash memory showed the excellent retention performance compared to the values reported previously. This performance is even comparable to those observed in the organic charge trapping memories, which typically showed better retention performance compared to organic floating-gate flash memories.

We revised the manuscript by as follows:

Page 16, 1st paragraph,

To the best of our knowledge, the flash memory device developed in this study exhibited one of the best retention characteristics as well as lowest $V_{\text{prg}}/V_{\text{ers}}$ among the organic FG flash memory based on polymer dielectrics, resulting from the tunneling-based programming mechanism through the robust dielectric layers with

high α_{CR} (Fig. **4c** and Supplementary Table **3**).⁶⁰⁻⁷⁰ Moreover, the flash memory showed the retention characteristics even superior or at least comparable to those obtained from the organic memories with different operating principles including charge trapping memories and photonic memories.⁷¹⁻⁷⁸ Accordingly, the 3D T-inverter fully maintained its three distinct logic states with full-swing operation over time owing to the excellent retention performance of the flash memory (Fig. **4d** and 4e).

71. Asadi K., De Leeuw D. M., De Boer B. & Blom P. W. Organic non-volatile memories from ferroelectric phase-separated blends. *Nat. Mater.* **7**, 547-550 (2008).

72. Koo J. et al. Nonvolatile electric double-layer transistor memory devices embedded with Au nanoparticles. *ACS Appl. Mater. & Interfaces* **10**, 9563-9570 (2018).

73. Zhou Y. et al. An upconverted photonic nonvolatile memory. *Nat. Commun.* **5**, 4720 (2014).

74. Chiu Y. C. et al. Oligosaccharide carbohydrate dielectrics toward high-performance non-volatile transistor memory devices. *Adv. Mater.* **27**, 6257-6264 (2015).

75. Wang W. et al. Highly reliable top-gated thin-film transistor memory with semiconducting, tunneling, charge-trapping, and blocking layers all of flexible polymers. *ACS Appl. Mater. & Interfaces* **7**, 10957-10965 (2015).

76. Tung W. Y. et al. High performance nonvolatile transistor memories utilizing functional polyimide-based supramolecular electrets. *Chem. Asian J.* **11**, 1631-1640 (2016).

77. Pak K., Choi J., Lee C. & Im S. G. Low-power, flexible nonvolatile organic transistor memory based on an ultrathin bilayer dielectric stack. *Adv. Electron. Mater.* **5**, 1800799 (2019).

78. Lee C. et al. Long-term retention of low-power, nonvolatile organic transistor memory based on ultrathin, trilayered dielectric containing charge trapping functionality. *Adv. Funct. Mater.* **30**, 2004665 (2020).

Reviewer (#2)'s COMMENTS:

In the manuscript, multi-valued logic inverter based on a heterojunction transistor and a nonvolatile floating-gate transistor were constructed with one heterojunction transistor stacked vertically on top of another nonvolatile floating-gate transistor. Intermediate V_{out} value of $\sim V_{DD}/2$, gain as high as 20 V/V were achieved, along with low driving voltage of less than 5 V. Moreover, long-term stability of the inverter was also obtained. This work is interesting and novel. It can be published in *Nat. Comm.* if the following comments can be properly addressed.

Response:

We appreciate the Reviewer #2's encouraging and constructive comments. We present the point-by-point response for each comment as follows.

Comment #1) In the experimental section, details for the inverter fabrication are missing, such as the thicknesses of the floating gate electrode and other relevant layers, and layer dimensions. These parameters are extremely important for others to reproduce the work.

Response:

We appreciate Reviewer #2's valuable comment. As Reviewer #2 suggested, we revised '**Device fabrication & characterization**' in Method section by adding the detailed information. Also, we corrected the channel dimension (800 μm of the width and 400 μm of the length).

We revised the manuscript by as follows:

Page 27, in Device fabrication & characterization section,

Device fabrication & characterization

To fabricate the vertically stacked organic 3D T-inverter, the flash memory device was fabricated. For gate and FG electrode, Al was thermally evaporated with the deposition rate of 0.1 nm/s. The thickness of the gate and FG electrode was commonly set to 50 nm, which was monitored *in-situ* by quartz crystal microbalance (QCM), and all the thermal evaporation processes were performed under high vacuum lower than 2×10^{-6} torr. The dielectric layers were deposited via iCVD process as

described above and the thicknesses of the pC1D1 BDL and pV3D3 TDL were 100 and 24 nm, respectively. The organic semiconductors including DNNT and PTCDI-C13 were also thermally deposited with the deposition rate of 0.03 nm/s. The PTCDI-C13 was recrystallized by the following thermal annealing at 200 °C for an hour. For *S/D* electrodes, 70 nm-thick Au was thermally evaporated with the deposition rate of 0.1 nm/s. The channel width and length were 800 and 400 μm, respectively. 1 μm-thick pV3D3 was deposited onto the flash memory by the iCVD process as an ILD, and the HTR was fabricated directly on top of the ILD. For the fabrication of the HTR, 50 nm-thick Al electrode was thermally evaporated and used as gate electrode and the thickness of pV3D3 dielectric layer was 40 nm. The partial PTCDI-C13 semiconducting layer was deposited and recrystallized at 200 °C for an hour, followed by the DNNT layer that ranges from *S* to *D* electrode. 70 nm-thick Au *S/D* electrode was deposited and the *D* electrode of the HTR was connected to *D* electrode of the flash memory through via-hole-less multi-metal interconnection. The overall areal dimension of the active area was 0.64 mm² (0.8 mm by 0.8 mm for width and length) including *S/D* electrodes, and the areal dimension of the patterned dielectric layer was 1.44 mm² (1.2 mm by 1.2 mm for width and length). To visualize the 3D T-inverter, the device was sliced by a focused ion beam (Helios Nanolab 450) and cross-sectional HRTEM images were taken by Cs-corrected TEM (Titan cubed G2, FEI) with EDS mapping analysis.

Comment #2) It is mentioned that a well-defined intermediate logic V_{out} state is important. How about the intermediate V_{in} value in T-inverters? Is the current work showing satisfactory intermediate V_{in} ?

Response:

We appreciate Reviewer #2's valuable comments. As Reviewer #2 pointed out, the V_{IN} value that corresponds to the intermediate logic state is also an important factor to determine the T-inverter performance.

In our device, the intermediate logic states were located at V_{IN} of 2.5 V with the pristine state and low V_{prg} including +14 and +16 V, however, the intermediate logic state was shifted toward higher V_{IN} range. This is because the NTC region was shifted to lower gate voltage ($|V_G|$) region with the decreasing $|V_D|$ as illustrated in Supplementary Fig. 16, which is mainly resulting from the reduced electron injection with the decreasing gate-to-drain voltage (V_{GD}). Therefore, with the optimum V_{prg} (+19 V), the intermediate logic value was shifted with the decreased $|V_D|$ of the HTR. In this regard, the optimum programming operation of the flash memory can force the I_D of the flash memory to match that of the HTR in the NTC region, which enabled the ideal V_{OUT} amplitude value (Supplementary Fig. 17). This ideal intermediate logic V_{IN} value could be achieved by the proposed strategy in this report where the flash memory was integrated into the T-inverter for the first time. In other words, the range of the intermediate logic V_{IN} state is determined mainly by the NTC region of a HTR, and we presented an effective way in this study to utilize the NTC behavior as an ideal V_{OUT} value in the intermediate logic state.

The length and position of the NTC region in the HTR directly determines the V_{IN} range and position of the intermediate logic state in the T-inverter, respectively. Therefore, systematic control of the length and position of NTC region in the HTR is highly required to achieve high-performance T-inverter. To achieve this goal, we are currently investigating two different strategies: (i) improving the charge transport characteristics of a n-type semiconductor to enhance the length of NTC region, and (ii) introducing an additional dielectric layer that can induce threshold voltage (V_T) shift to control the position of the NTC region (**Fig. R1**).

Figure R1. Schematic illustration of the change in transfer characteristics of a HTR according to the **a**, improving n-type semiconductor performance and **b**, introducing V_T controlling layer.

As a preliminary experiment to verify the validity of the strategy (i), we increased the thickness of n-type PTCDI-C13 partial semiconductor ($d_{\text{PTCDI-C13}}$) to facilitate the free carrier density and charge transport (*Appl. Phys. Lett.* **2011**, 98, 073307 and *Org. Electron.* **2010**, 11, 1920-1927). With the increasing thickness of n-type semiconductor, the length of NTC region was improved from 0.8 to 1.1 V (**Fig. R2**), which showed feasibility of the strategy (i).

Figure R2. The device structure of the HTR employing n-type PTCDI-C13 semiconductor with thickness of **a**, 30 and **b**, 45 nm. **c**, The transfer characteristics of the HTR according to the PTCDI-C13 thickness and **d**, V_{peak} , V_{valley} extracted from the transfer curves.

Also, we introduced the V_T control layer to verify that the position of NTC region can be controlled (strategy (ii)). In this experiment, the thicknesses of both semiconductors were commonly fixed to 30 nm. As shown in **Fig. R3a-d**, the NTC region was shifted toward higher $|V_G|$ after employing the V_T control layer without hampering the length of NTC region. Moreover, the ideal V_{OUT} value in the intermediate logic state was achieved at the V_{IN} of 2.5 V (**Fig. R3e-f**). Please note that the flash memory with the optimum programming state ($V_{prg}=+17$ V) was still very necessary to achieve the I_D balance between HTR and a complementary transistor (**Fig. R3e**).

Figure R3. The device structure of **a**, control HTR and **b**, HTR with a V_T control layer. **c**, The transfer characteristics of the HTR with respect to the presence of V_T control layer and **d**, V_{peak} , V_{valley} extracted from the transfer curves. **e**, the overlapped transfer curves of HTR employing

the V_T control layer and flash memory with the optimum programming state ($V_{\text{prg}}=+17$ V). **f**, The voltage transfer curve (VTC) of the T-inverter composed of the HTR employing V_T control layer and flash memory with the optimum programming state ($V_{\text{prg}}=+17$ V).

Even though we could accomplish the matching between V_{IN} and V_{OUT} values in the intermediate logic state in our preliminary experimental results as shown above, it is still highly required to enlarge the V_{IN} range where the intermediate V_{OUT} value appeared, to develop the high-performance T-inverter with the maximized noise margin. To achieve this goal, the position of the NTC region should further be optimized and its range should be enlarged by combining the strategy (i) and (ii) described above. As we verified that the length of NTC region can be increased with the improved electrical characteristics of n-type partial semiconductor (**Fig. R2**), it can be replaced to other high-performance n-type semiconductor such as two-dimensional molybdenum disulfide (2D MoS₂) and amorphous indium-gallium-zinc oxide (a-IGZO) for further improvement. Also, employing functional dielectric layer that can control the V_T of n-type OTFT or high- k dielectric layers developed in our previous reports (*Adv. Funct. Mater.* **2016**, *26*, 6574-6582, *ACS Appl. Mater. Interfaces* **2017**, *9*, 20808-20817 and *Adv. Electron. Mater.* **2020**, *6*, 200314) can allow us to control the position of the NTC region.

Together with the optimization of the NTC behavior of the HTR, it is of critical importance to introduce the floating-gate memory structure into the complementary transistor as we demonstrated in this study, in order to match the channel conductance of the HTR and complementary transistor and to utilize the NTC region of the HTR as an ideal intermediate logic state. Considering all the aspects described above, we are indeed currently working on the optimization of the NTC behavior of the HTR and corresponding T-inverter performance with appropriate design/simulation approach. However, we believe these strategies to control of NTC behavior are beyond the scope of this report, and we hope that we can show more convincing result in near future as a separate report with systematic study.

We revised the manuscript by adding the transistor characteristics of HTR with different V_D and the overlapped transfer curves of the flash memory and HTR to analyze the T-inverter characteristics:

Supplementary Figure 16 | **a**, The transfer characteristics of the HTR with different V_D . **b**, the change in V_{peak} and V_{valley} with respect to V_D .

Supplementary Figure 17 | The overlapped transfer characteristics of the flash memory according to V_{prg} and the HTR with different V_D ; **a**, $V_{\text{prg}} = +14$ V, $V_{D,\text{HTR}} = -4$ V, **b**, $V_{\text{prg}} = +16$ V, $V_{D,\text{HTR}} = -3.5$ V, **c**, $V_{\text{prg}} = +18$ V, $V_{D,\text{HTR}} = -3$ V and **d**, $V_{\text{prg}} = +19$ V, $V_{D,\text{HTR}} = -2.5$ V. **e-h**, The corresponding VTC of the T-inverter.

A discussion was added accordingly:

Page 13, 1st paragraph,

The intermediate logic value was gradually increased in accordance with the increasing V_{prg} and reached to 2.49 V with $V_{\text{prg}}=+19$ V (Fig. **3g**). Therefore, the optimum programming state of the flash memory allowed that the intermediate logic value is practically identical to $V_{\text{DD}}/2$ (logic "1"), which made the intermediate logic state clearly distinguishable from V_{DD} (logic "2") and G_{ND} (logic "0"). The V_{IN} range where the intermediate logic states appeared was gradually shifted with the increasing V_{prg} , because the NTC region of the HTR was shifted to lower $|V_{\text{G}}|$ region with the decreasing $|V_{\text{D}}|$ (Supplementary Fig. **16**). The V_{IN} range of the intermediate logic states decreased slightly with the increasing V_{prg} , because only the NTC region in the HTR could be represented to the intermediate logic state, which enabled us to obtain the ideal V_{out} value ($V_{\text{DD}}/2$) by optimizing the memory programming process (Supplementary Fig. **17**). The 1st gain decreased gradually while 2nd gain increased with the increasing V_{prg} , resulting from the increased intermediate logic value (Fig. **3h** and Supplementary Fig. **18**).

Page 18 in Supplementary Information,

The transfer characteristics of the HTR with different V_{D}

To investigate the origin that causes the shift of the intermediate logic state, we measured the transfer characteristics of the HTR with different V_{D} , because V_{D} of the HTR is determined by the difference between the supply voltage and the intermediate logic value (Supplementary Fig. **16a**). With the decreasing $|V_{\text{D}}|$ of the HTR, the NTC region was shifted toward lower $|V_{\text{G}}|$, which led the shift of the intermediate logic state to higher V_{IN} . Nevertheless, the length of the NTC region was fully preserved (Supplementary Fig. **16b**) The shift of the NTC region is most likely resulting from the reduced electron injection with the decreasing gate-to-drain voltage (V_{GD}) so that the depletion of n-type PTCDI-C13 started to occur at lower $|V_{\text{G}}|$.

Page 19 in Supplementary Information,

The overlapped transfer curves of the transistors and resulting VTC of the T-inverter according to memory programming state

Supplementary Fig. **17a-d** shows the overlapped transfer curves of the flash memory and HTR to analyze the intermediate logic state according to the memory programming state. The VTCs of the T-inverter was also shown in Supplementary Fig. **17e-h** along with each memory programming state. With the low V_{prg} (+14 and +16 V), the V_G region where the flash memory and HTR exhibited similar I_D (less than an order of magnitude difference) was relatively wide (~ 1.2 V) (Supplementary Fig. **17a, b**). In those V_G regions, the flash memory showed slightly higher I_D compared to that of the HTR, resulting in low intermediate logic value (1.09 and 1.47 V at $V_{prg} = +14$ and +16 V, respectively). However, with the V_{prg} of +19 V, both transistors showed quite similar I_D values in the NTC of the HTR, which produced the ideal V_{OUT} value ($\sim V_{DD}/2$) in the intermediate logic state. In this optimum programming state, the V_{IN} range of the intermediate logic state decreased slightly to ~ 0.9 V, because only the NTC region in the HTR can represent the intermediate logic state.

Comment #3) The hysteresis of a transistor and inverter are highly dependent on the sweeping speed of the input voltage. It is not accurate to say this transistor/inverter is hysteresis-free without mentioning the sweeping speeds.

Response:

We appreciate Reviewer #2 for the constructive comment. We measured the transfer characteristics of the transistors and VTC of the inverter with different sweeping speed, which verified that no notable hysteresis behavior was obtained regardless of the sweeping speed ranging from 0.47 to 0.03 V/s, as shown in Supplementary Fig. **10** and **11**. We added the sweeping speed information in the revised manuscript.

We revised the manuscript by as follows:

Page 11, 2nd paragraph,

All the fabricated 3D T-inverters showed full-swing operation with the uniform electrical characteristics stemming from the excellent device-to-device uniformity of the constituent devices, together with the high I_{on}/I_{off} ratio of the HTR, which can be hardly achieved in conventional AATs where two semiconductors are overlapped only at the center of the channel.^{27, 28} Also, no notable hysteresis was observed in the transfer curves of the flash memory and HTR, and the VTC of the T-inverter regardless of the sweeping speed (Supplementary Fig. **10** and **11**). Moreover, all the 3D T-inverters displayed a distinct intermediate logic state with two clearly distinguishable maximum gain values.

Page 28, in **Device fabrication & characterization section**,

The electrical characteristics were measured by B1500A semiconductor analyzer (Agilent Technologies). The sweeping speed of the input voltage was set to ~ 0.1 V/s, otherwise specified. The temperature of the devices was measured by infrared (IR) thermometer (Fluke 62 MAX). All the device fabrication and characterization were performed in the N₂-filled glovebox.

We added transfer curves of the transistors and VTC of the inverter with different sweeping speed.

Supplementary Figure 10 | The transfer curves of the **a**, flash memory and **b**, HTR with different sweeping speed.

Supplementary Figure 11 | The VTCs of the T-inverter with different sweeping speed.

Discussion was added accordingly:

Page 12 in Supplementary Information,

The electrical characteristics of the transistors and inverter with different measuring speed.

The transfer characteristics of the flash memory and HTR measured with each gate voltage (V_G) sweeping speed were shown in Supplementary Fig. **10**. The VTCs of the T-inverter with each input voltage (V_{IN}) sweeping speed were also shown in Supplementary Fig. **11**. No notable hysteresis was observed, and extremely low gate leakage current was fully maintained regardless of the sweeping speed in the transfer curves of the transistors. The T-inverter also showed only negligible amount of hysteresis with the sweeping speed as low as 0.06 V/s.

Comment #4) Related to the last comments, how fast the inverter can be switched? What is the delay time? This is one important parameter for an inverter.

Response:

We appreciate Reviewer #2 for the constructive comment. As Reviewer #2 suggested, we measured the switching speed values of the T-inverter. A switching speed within a few hundreds of milliseconds was obtained, which is comparable to those obtained generally in the organic binary logic circuits, which showed that the switching speed was not degraded with the increasing number of logic states. The switching speed of the OTFT is closely related to the channel length, charge mobility and contact resistance (*Org. Electron.* **2017**, *49*, 179-186, *Adv. Funct. Mater.* **2020**, *30*, 1909501 and *Sci. Adv.* **2020**, *6*, eaaz5156). Therefore, the switching speed of our device can further be improved by optimizing the channel dimension and fabrication process.

We revised the manuscript by adding switching speed information:

Supplementary Figure 21 | **a**, The applied V_{IN} and **b**, V_{OUT} in response to the applied V_{IN} of the T-inverter with the optimum programming state.

We also added the summary of the measured switching speed:

Supplementary Table 2 | Summary of the parameters for the switching speed of the T-inverter.

Switching number	V_{OUT} (V)		Time (s)		Switching speed (ms)
	initial	terminal	initial	terminal	
1	0	2.6	0.60	0.82	220
2	2.6	5	1.19	1.31	120
3	5	2.6	1.80	2.09	290
4	0	5	2.99	3.12	130
5	5	0	3.60	3.91	310
6	0	2.6	4.19	4.41	220
7	2.6	5	4.81	5.12	310

Discussion was added accordingly:

Page 14, 2nd paragraph,

The 3D T-inverter with both states exhibited no notable hysteresis regardless of the V_{IN} history throughout the measuring time. However, the intermediate logic value was clearly enhanced in the programming state, compared to that observed in the pristine state. The switching speed of the T-inverter was also investigated (Supplementary Fig. **21** and Supplementary Table **2**), which showed that the switching speed was comparable to those obtained frequently in the organic binary logic circuits.⁵⁷⁻⁵⁹ It follows from the above observation that the electrical characteristics of the vertically stacked 3D T-inverter, particularly the intermediate logic state, could be controlled systematically by the programming/erasing states of the flash memory.

57. Zschieschang U., Bader V. P. & Klauk H. Below-one-volt organic thin-film transistors with large on/off current ratios. *Org. Electron.* **49**, 179-186 (2017).

58. Chang J. S., Facchetti A. F. & Reuss R. A circuits and systems perspective of organic/printed electronics: review, challenges, and contemporary and emerging design approaches. *IEEE J. Emerg. Sel. Top.* **7**, 7-26 (2017).

59. Kumar B., Kaushik B. K. & Negi Y. S. Organic thin film transistors: structures, models, materials, fabrication, and applications: a review. *Polym. Rev.* **54**, 33-111 (2014).

Page 25 in Supplementary Information,

The analysis on the switching speed of the T-inverter

To investigate the switching speed of the 3D T-inverter with the optimum programming state, we measured the V_{OUT} in the response to the applied V_{IN} according to time (Supplementary Fig. **21** and Supplementary Table **2**). The switching speed of the T-inverter was calculated as the difference between the time when a 10% increase in V_{OUT} occurs and the time when a 90% of the terminal V_{OUT} was achieved. A switching speed within a few hundreds of milliseconds was obtained, which is comparable to those obtained generally in the organic binary logic circuits.¹⁰⁻¹²

10. Zschieschang U., Bader V. P. & Klauk H. Below-one-volt organic thin-film transistors with large on/off current ratios. *Org. Electron.* **49**, 179-186 (2017).
11. Chang J. S., Facchetti A. F. & Reuss R. A circuits and systems perspective of organic/printed electronics: review, challenges, and contemporary and emerging design approaches. *IEEE J. Emerg. Sel. Top.* **7**, 7-26 (2017).
12. Kumar B., Kaushik B. K. & Negi Y. S. Organic thin film transistors: structures, models, materials, fabrication, and applications: a review. *Polym. Rev.* **54**, 33-111 (2014).

Comment #5) It is suggested to plot/draw the illustration of the inverter/transistor with both side view and top view, so that the structure of the devices can be much clearer.

Response:

We appreciate Reviewer #2 for the valuable comment. As Reviewer #2 suggested, we have added the 3D images to clearly show the structure of the devices.

We revised the manuscript by adding the 3D images of the device:

Supplementary Figure 1 | **a**, A schematic of the device and **b**, Microscopic image and schematic illustration of the device with different angles according to the fabrication procedure (top: organic flash memory in the 1st floor, bottom: organic HTR in the 2nd floor).

Discussion was added accordingly:

Page 6, 2nd paragraph,

The schematic illustration for the fabrication process and the structure of the vertically stacked organic ternary logic inverter are shown in Fig. **1a** and **b**, respectively. All the dielectric layers were deposited by iCVD process and patterned *in-situ* through shadow mask to achieve via-hole-less metal interconnection.¹⁶ Also, a 1 μm -thick interlayer dielectric (ILD) was imposed between the unit devices to electrically isolate the flash memory and HTR. Schematic illustration of the device structure with various view points according to the fabrication process is shown in Supplementary Fig. **1**. Organic flash memory including FG was introduced to match the channel conductance

with the HTR in the NTC region, thus to optimize the voltage transfer characteristic (VTC) and intermediate logic state of the resulting 3D T-inverter (Fig. **1c** and **d**).

Page 2 in Supplementary Information,

Schematic illustration of the device

The schematic illustration of the device is shown in Supplementary Fig. **1**. Note that the gate electrode of the flash memory and HTR was connected and the drain electrode of the flash memory and HTR was connected through the via-hole-less metal interconnection (Supplementary Fig. **1a**). Supplementary Fig. **1b** shows the optical microscopy image of the fabricated device and corresponding schematic illustration of the device according to the fabrication procedure.

Reviewer (#3)'s COMMENTS:

Excellent paper, very well structured, novel and interesting. The authors incorporate several novel ideas, including the use of new CVD-deposited insulators, and the use of these materials to create a flash memory. The paper further includes a comprehensive characterization of the fabricated devices and uses more than enough devices and sweep measurements to confirm a scalable and reliable process. The paper also includes a reasonable background and cites the background work of other groups well.

I recommend for acceptance without any major modifications necessary.

Response:

We appreciate the positive and encouraging comments from Reviewer #3.

We very appreciate the Reviewers' all the valuable comments, which greatly helped us to improve this manuscript.

Reviewer comments, second round -

Reviewer #1 (Remarks to the Author):

The author had addressed most comments from the first round review. It can be published now in Nature Communications.

Reviewer #2 (Remarks to the Author):

The authors took into consideration my comments and addressed them in full. Thus, I believe that the paper can be published as it is

Response to the Reviewers' Comments

The authors thank the Reviewers for their considerate review of our manuscript and the valuable comments. We have revised the manuscript and supporting information based on the Reviewers' comments. The Reviewers' comments appear in **black**, and the authors' responses in blue. In the revised manuscript, **the changes** with respect to the previous version are highlighted in yellow.

Reviewer (#1)'s COMMENTS:

This paper proposed an interesting solution to achieve multi-valued logic circuits via a combination of heterojunction transistor (HTR) and flash memory and fabricated a 3-dimensional (3D) T-inverter with a vertically stacked form based on all-dry processes. This work is interesting. Though the authors claimed outstanding device uniformity and stable performance were realized, the significance and the novelty of this work are not clear. For example, compared with ref 18 and other previous work, what are the technical advantages in the device fabrication? Further, what is the point to use a T-inverter for multi-valued logic circuits? The gain of such a device is generally lower compared with conventional binary inverters. The question is how to balance the number of logic values and the gain for each value? This paper now is not suitable for publication in *Nature Communications*, though a major revision is recommended.

Response:

We appreciate the Reviewer #1 for in-depth review of our manuscript and providing the valuable comments. We have carefully revised the manuscript by preparing the point-by-point response to each comment as follows.

General Comment #1) Though the authors claimed outstanding device uniformity and stable performance were realized, the significance and the novelty of this work are not clear. For example, compared with ref 18 and other previous works, what are the technical advantages in the device fabrication?

Response:

As the Reviewer #1 pointed out, our group previously demonstrated organic logic circuits by using via-hole-less metal interconnection to achieve high-density organic electronics (*IEEE Electron Device Lett.* **2020**, *41*, 1685-1687 (Ref. 18 in the manuscript) and *Nat. Commun.* **2019**, *10*, 2424). We would like to highlight the technical advances in the device fabrication achieved in this paper compared to our previous reports, as summarized in **Table R1**.

Table R1. Summary of the technical advances in organic circuits based on via-hole-less metal interconnection

	Nat. Commun. 2019.	IEEE Electron Device Lett. 2020.	This work
Unit device	Transistor (p, n)	Transistor (p, n)	Heterojunction transistor (p) Floating-gate flash memory (n)
Circuit configuration	Binary circuit	Binary circuit	Ternary logic circuit
Circuit programmability	X	X	O
Vertical integration	O	X	O
Uniformity (Number of devices)	20	12	22
Operating voltage	10 V	10 V	5 V

The primary technical advance in this report is the expansion of the applicability of the via-hole-less metal interconnection scheme into multi-valued logic circuits. Compared to the previous reports where conventional digital logic circuits were developed, we newly designed and demonstrated a ternary logic circuit in vertically stacked structure.

Especially in this study, an organic nonvolatile flash memory instead of a transistor was newly incorporated into the ternary logic inverter (T-inverter), which is, to the best of our knowledge, the first demonstration of the flash memory-integrated ternary logic circuit. The

incorporation of a nonvolatile flash memory allows for the systematic circuit programmability, where the precise control of the intermediate logic state in the T-inverter is enabled. With the new circuit design, the fabricated devices in this study clearly showed outstanding device performance, featured by well-defined three discernable logic states, full-swing operation as well as highly stable, hysteresis-free operation, which had not been achieved to date. Meanwhile, the ternary logic devices demonstrated in this report fully maintained the important advantages obtained in the previous binary logic circuits fabricated by using via-hole-less metal interconnection, such as the excellent device-to-device uniformity and low operating voltage (less than 5 V).

In addition, to achieve low programming/erasing voltage ($V_{\text{prg}}/V_{\text{ers}}$) of the flash memory as well as low operating voltage, we introduced a high- k dielectric layer, poly(2-cyanoethyl acrylate-*co*-diethylene glycol divinyl ether) (pCEA-*co*-DEGDVE) as a blocking dielectric layer (BDL) and verified its compatibility with the via-hole-less multi-metal interconnection architecture. Also, we employed an ultrathin tunneling dielectric layer (TDL) with via-hole-less metal interconnection, whose thickness was less than 30 nm, about a half of the dielectric thickness used in our previous reports.

Considering all these aspects, we believe the significance and novelty of this work can be justified.

We revised the manuscript as follows

Page 5, 2nd paragraph,

As an effective method to systematically control the electrical characteristics (i.e. channel conductance) of a transistor, a nonvolatile floating-gate (FG) flash memory was utilized and integrated with HTR to implement a T-inverter in this study. The implementation of the nonvolatile FG flash memory enabled a systematic circuit programmability, where the precise control of the threshold voltage (V_t) allows to enhance the T-inverter performance by optimizing the intermediate logic state. Furthermore, by utilizing the via-hole-less metal interconnection method,^{16, 18} the flash memory on the first floor was vertically integrated with the HTR on the second floor to realize a three dimensional (3D) T-inverter, which is, to the best of our knowledge, the first demonstration of a vertically stacked MVL circuit.

Page 6, 2nd paragraph,

In the 3D T-inverter fabrication process, a vapor-phase deposition method, termed initiated chemical vapor deposition (iCVD) process was introduced to deposit high-performance polymer dielectric layers while minimizing the damage to the underlying devices.⁴⁰⁻⁴³ The polymer dielectric materials were introduced with optimized dielectric layer configuration, which reduced the programming/erasing voltage ($V_{\text{prg}}/V_{\text{ers}}$), and enabled the operating voltage less than 5 V, which is comparable to or even lower than those obtained in the previous organic 3D logic circuits.¹⁴⁻¹⁶ Meanwhile, the excellent device-to-device uniformity was also fully retained. Most of all, the solvent-free deposition process was capable of in-situ patterning of the dielectric layer simply through shadow mask during the deposition process, which allowed etching-free, via-hole-less metal interconnection, and thus 3D vertical integration of the HTR and flash memory devices.¹⁶

Page 17, 3rd paragraph,

In summary, we demonstrated a low-voltage organic ternary logic circuit, in which the organic HTR was vertically integrated with the organic nonvolatile flash memory. The integration of a nonvolatile flash memory into T-inverter allowed for the circuit programmability, where the electrical characteristics including intermediate logic state can be controlled precisely by charge storage in FG. All the device fabrication processes were based on solvent-free vapor-phase deposition methods, which enables the high degree of uniformity in the device performance. By applying the via-hole-less, vertical interconnection scheme into the MVL circuit, the T-inverter in 3D structure was fabricated for the first time, which can enhance the device density per unit area. In the organic flash memory, the high- k and low- k polymer dielectric pairs were used as BDL and TDL, respectively, which could effectively reduce the $V_{\text{prg}}/V_{\text{ers}}$ as well as operating voltage (~ 5 V).

General Comment #2) Further, what is the point to use a T-inverter for multi-valued logic circuits? The gain of such a device is generally lower compared with conventional binary inverters. The question is how to balance the number of logic values and the gain for each value?

Response:

As Reviewer #1 commented, the gain value of the T-inverters can be inherently low compared to that of conventional binary inverters due to two different switching occurring in T-inverters. Nevertheless, ternary logic circuits based on HTRs exhibit a great advantage that the multi-logic circuits can drastically reduce the system complexity (*Nat. Nanotechnol.* **2017**, *12*, 1148-1154, *Nat. Mater.* **2017**, *16*, 170-181, *Nat. Commun.* **2019**, *10*, 1998 and *Adv. Sci.* **2021**, *8*, 2004216). Furthermore, please note that the DC gain is critically important in amplifier and high-speed buffer rather than arithmetic circuits as bandwidth in amplifier circuit and high-pass cutoff frequency are obtained by the degree of DC gain (*J. Low Power Electron. Appl.* **2019**, *9*, 26).

Moreover, compared to the previously reported ternary logic circuits, the gain values here can be regarded as one of the highest achieved to date, as summarized in **Table R2**. This information can also be found in **Supplementary Table 1**.

Table R2. Summary of the operating voltage and DC gain values in this study compared to the previous ternary logic circuits.

Reference	V_{IN}/V_{OUT} (V)	V_{DD} (V)	DC gain (V/V)
Nat. Commun. 2019 , 10 , No. 1998	5 / 5	5	>1
Nano Lett. 2018 , 18 , 4355-4359	4 / 8	10	N/A
Adv. Mater. 2019 , 31 , 1808265	50 / 50	50	>20
Nano Lett. 2016 , 16 , 1359-1366	1 / 0.9	1	4.5
Nat. Commun. 2016 , 7 , No. 13413	25 / 1.7	2	N/A
Nat. Nanotechnol. 2017 , 12 , 1148-1154	1 / 2	2	12
ACS Nano 2019 , 13 , 4478-4485	60 / 20	26	1.2
ACS Nano 2019 , 13 , 5430-5438	40 / 2	2	0.2

ACS Appl. Mater. & Interfaces 2020 , 12 , 36530-36539	80 / 8	8	N/A
Semicond. Sci. Technol. 2020 , 35 , 065020	80 / 10	10	1
Adv. Electron. Mater. 2020 , 6 , 2000426	2 / 2	2	N/A
This work	5 / 5	5	~20

Only one study reported the gain value comparable with this study (*Adv. Mater.* **2019**, *31*, 1808265). However, the operating voltage in that study was as high as 50 V, which is ten times higher than that achieved in this study, mainly due to the use of thick, rigid SiO₂ dielectric layer. In addition, we believe this paper can provide an important insight to optimize the T-inverter performance. For example, the 2nd gain value was dramatically enhanced from -3 to -19 V/V with the increased intermediate logic value by programming the n-type flash memory.

Even though ternary logic circuits are suitable to arithmetic circuit applications rather than amplifier or high-speed buffer as discussed above, the DC gain values obtained in this study (~20 V/V) are not quite low and comparable to many gain values achieved in the organic binary logic circuits with polymer dielectrics reported recently (*IEEE Trans. Electron Devices* **2014**, *61*, 1175-1180, *IEEE Trans. Electron Devices* **2014**, *61*, 2220-2223, *Adv. Electron. Mater.* **2017**, *3*, 1600557, *Org. Electron.* **2017**, *46*, 14-21, *Adv. Electron. Mater.* **2018**, *4*, 1700313, *Adv. Electron. Mater.* **2018**, *4*, 1800340, *Org. Electron.* **2019**, *75*, 105358, *J. Mater. Chem. C* **2020**, *8*, 15331-15338 and *ACS Appl. Mater. Interfaces* **2021**, *13*, 30921-30929) (Ref. 48-56 in the revised manuscript), where in most cases, a few tens of the DC gain values were obtained.

Considering the Reviewer #1's General Comment #2, we revised the manuscript by as follows:

Page 13, 1st paragraph,

Nevertheless, the obtained gain value was one of the highest DC gain values among the previously reported non-Si-based ternary logic inverters reported to date (Supplementary Table **1**).^{23, 29, 30, 32-34, 37} Also, the obtained gain values were

comparable to those observed in the organic binary logic inverters with polymer dielectric materials.⁴⁸⁻⁵⁶ Moreover, the dramatic increase of the 2nd gain value strongly suggests that the balance between channel conductance of the flash memory and HTR is important to achieve high-performance T-inverter, which was achieved successfully in this study by replacing the n-type transistor with a flash memory.

48. Feng L. et al. All-solution-processed low-voltage organic thin-film transistor inverter on plastic substrate. *IEEE Trans. Electron Devices* **61**, 1175-1180 (2014).

49. Feng L., Cui Q., Zhao J., Tang W. & Guo X. Dual-V_{th} low-voltage solution processed organic thin-film transistors with a thick polymer dielectric layer. *IEEE Trans. Electron Devices* **61**, 2220-2223 (2014).

50. Shiwaku R. et al. Printed organic inverter circuits with ultralow operating voltages. *Adv. Electron. Mater.* **3**, 1600557 (2017).

51. Shin E.-Y., Choi E.-Y. & Noh Y.-Y. Parylene based bilayer flexible gate dielectric layer for top-gated organic field-effect transistors. *Org. Electron.* **46**, 14-21 (2017).

52. Takeda Y. et al. Organic complementary inverter circuits fabricated with reverse offset printing. *Adv. Electron. Mater.* **4**, 1700313 (2018).

53. Stucchi E., Dell'Erba G., Colpani P., Kim Y. H. & Caironi M. Low-voltage, printed, all-polymer integrated circuits employing a low-leakage and high-yield polymer dielectric. *Adv. Electron. Mater.* **4**, 1800340 (2018).

54. Jo I. Y. et al. Low-voltage-operating complementary-like circuits using ambipolar organic-inorganic hybrid thin-film transistors with solid-state-electrolyte gate insulator. *Org. Electron.* **75**, 105358 (2019).

55. Stucchi E., Scaccabarozzi A. D., Viola F. A. & Caironi M. Ultraflexible all-organic complementary transistors and inverters based on printed polymers. *J. Mater. Chem. C* **8**, 15331-15338 (2020).

56. Park H. et al. Tailored polymer gate dielectric engineering to optimize flexible organic field-effect transistors and complementary integrated circuits. *ACS Appl. Mater. & Interfaces* **13**, 30921-30929 (2021).

Comment #1) On page 6, line 131: “a 1 μm -thick interlayer dielectric (ILD) was imposed between the unit devices to electrically isolate the flash memory and HTR.”, will this interlayer hamper the heat dissipation of the underlying device and affect its stability? The author should track the temperature of the device during continuous operation.

Response:

We appreciate the Reviewer #1 for the important comments. To clarify the heat dissipation of the underlying floating-gate device, we carried out a two-dimensional (2D) finite-element simulation by reproducing the fabricated devices including 1 μm -thick polymeric interlayer dielectric (ILD). The simulation result revealed that the temperature rise was within a level of 0.1 K, despite the low thermal conductivity of the organic materials employed for the device and the use of a relatively thick ILD polymer on the flash memory device. Such an observation is mainly due to the low operating voltage (~ 5 V).

Moreover, as Reviewer #1 suggested, thermal analysis was also performed experimentally by stacking 1 μm -thick pV3D3 ILD on top of the fabricated PTCDI-C13 flash memory. We measured the temperature using infrared (IR) thermometer under the continuous operation at $V_G = V_D = 5$ V and only the negligible temperature variation was observed throughout the continuous operation up to 5 hours.

We revised the manuscript by adding thermal analysis results as below:

Supplementary Figure 5 | Simulated physical quantities inside a PTCDI-C13 transistor at $V_G=V_D=5$ V. **a**, electron concentration, **b**, Joule heat power, and **c**, lattice temperature. **d**, Dependence of the maximum lattice temperature on the driving voltages.

Supplementary Figure 6 | **a**, Photography for the measurement of the device temperature with an hour interval. **b**, The change in I_D and temperature of PTCDI-C13 flash memory with 1 μ m-thick ILD during continuous operation at $V_G=V_D=5$ V.

A discussion on this aspect was added accordingly:

Page 9, 1st paragraph,

Furthermore, the average threshold voltage (V_T) was less than 0.6 V with narrow distribution (0.55 ± 0.09 V), thus ensuring the uniform channel conductance of the pull-down transistor. **It is also worthwhile to note that even with a relatively thick polymer ILD with low thermal conductivity, only negligible change in the temperature**

of the flash memory was observed during the continuous operation at 5 V, owing to the low operating voltage (Supplementary Fig. 5 and 6).

Details of the temperature measurement was added in Method section:

Page 28, in **Device fabrication & characterization** section,

The electrical characteristics were measured by B1500A semiconductor analyzer (Agilent Technologies). The sweeping speed of the input voltage was set to ~ 0.1 V/s, otherwise specified. The temperature of the devices was measured by infrared (IR) thermometer (Fluke 62 MAX). All the device fabrication and characterization were performed in the N_2 -filled glovebox.

More detailed discussion for thermal analysis result and the related references were added in Supplementary Information:

Page 7 in Supplementary Information,

Thermal analysis of the flash memory under polymer ILD

To clarify the heat dissipation of the underlying floating-gate device, we performed a two-dimensional finite-element simulation to investigate the thermal properties of the proposed system.⁵ The thermal boundary condition was set to 300 K at metal electrodes and the thermal conductivity and heat capacity of all organic materials (PTCDI-C13, pV3D3, pC1D1) was assumed to be 10^{-3} $Wcm^{-1}K^{-1}$ and 1 $Jcm^{-3}K^{-1}$, respectively.⁶ Supplementary Fig. 5a-c present the simulated physical quantities inside the PTCDI-C13 thin film at $V_G = V_D = 5$ V. A strong accumulation of electrons at the PTCDI-C13/pV3D3(TDL) interface is clearly identifiable in Supplementary Fig. 5a with an apparent channel pinch-off at the drain. This in turn resulted in pronounced Joule heating over the pinch-off region due to an elevated electric field, as shown in Supplementary Fig. 5b. Supplementary Fig. 5c shows the internal temperature distribution, where the direction and degree of heat dissipation from the thermal hot spot is distinctively visualized. Most importantly, despite the low thermal conductivity of organic materials and the use of a relatively thick interlayer dielectric polymer film

on top of the device, the temperature rise was practically negligible – no larger than 0.1 K with the low operating voltage (5 V), as illustrated in Supplementary Fig. **5d**.

Moreover, the temperature of the PTCDI-C13 flash memory fabricated with 1 μ m-thick pV3D3 ILD was measured under the continuous operation at $V_G = V_D = 5$ V. As shown in Supplementary Fig. **6**, the temperature change less than 0.1 K was observed throughout the whole continuous operation up to 5 hours, which is fully consistent with the result obtained by the two-dimensional finite-element simulation.

5. Han H. & Kim C.-H. Unexpected Benefits of Contact Resistance in 3D Organic Complementary Inverters. *Adv. Electron. Mater.* **6**, 1900879 (2020).

6. Wang, H. & Yu, C. Organic Thermoelectrics: Materials Preparation, Performance Optimization, and Device Integration. *Joule* **3**, 53-80 (2019).

Comment #2) Please carefully check the statement from line 137 to 140 on page 7.

Response:

We appreciate Reviewer #1's attentive review. We also rechecked the whole manuscript thoroughly and corrected all errors and typos as much as we can (including Response to Comment #3).

We revised the manuscript as follows:

Page 7, 1st paragraph,

Note that the drain electrode on top of PTCDI-C13/DNTT heterojunction in HTR was interconnected to the drain electrode in the flash memory and used as an output electrode in the inverter operation to induce an intermediate logic state.²⁹

Page 27, in **Device fabrication & characterization** section,

Device fabrication & characterization

For *S/D* electrodes, 70 nm-thick Au was thermally evaporated with the deposition rate of 0.1 nm/s. The channel width and length were 800 and 400 μm , respectively. 1 μm -thick pV3D3 was deposited onto the flash memory as an ILD and the HTR was fabricated directly on top of the ILD.

The scale bars in the optical microscope (OM) images were corrected accordingly:

Page 9, in Supplementary Information,

Supplementary Figure 7 | The OM images (left) and AFM images (right) at the middle of the active layer regions of **a**, the flash memory, **b**, ILD deposited on top of the flash memory and **c**, HTR.

Comment #3) On page 9, line 188 to 189, the author claimed that all on/off ratios are higher than 10^6 , however, in figure 2c, it seems most of them are lower than 10^6 , please double-check this part.

Response:

We appreciate Reviewer #1's critical review comments. We rechecked the raw data and found that the on/off ratio was slightly lower than 10^6 . We revised the manuscript accordingly. The on/off current value at each current level and the calculated on/off current ratio (I_{on}/I_{off}) therefrom are also summarized in **Table R4**.

Table R4. The I_{off} , I_{on} and I_{on}/I_{off} values of the organic flash memory

Device #	1	2	3	4	5	6	7	8	9	10	11
I_{off} ($\times 10^{-13}$ A)	2.63	4.47	3.37	3.08	3.39	3.60	2.25	3.24	4.20	3.08	2.30
I_{on} ($\times 10^{-7}$ A)	1.97	2.02	1.99	2.01	1.97	1.90	1.93	1.90	1.97	1.90	1.98
I_{on}/I_{off} ($\times 10^5$)	7.50	4.51	5.89	6.51	5.80	5.27	8.58	5.87	4.70	6.17	8.63

We revised the manuscript as follows:

Page 9, 1st paragraph,

As shown in the statistical analysis, all the flash memory devices exhibited on/off current ratio (I_{on}/I_{off}) higher than 10^5 with the operating voltage less than 5 V (Fig. **2c**).

Comment #4) On page 10, line 206, please explain the BTBT in detail.

Response:

We appreciate Reviewer #1 for the constructive suggestion. To elucidate the BTBT in more detail, we added a discussion on the operating principle of the HTR by dividing the operation states in the transfer curve into three regimes.

We revised the manuscript as follows:

Page 10, 1st paragraph,

All the HTR exhibited NTC characteristics with four different operation regions: (i) off-state ($-0.8 \text{ V} < V_G < 2 \text{ V}$), (ii) subthreshold region ($-1.5 \text{ V} (= V_{\text{peak}}, \text{ peak voltage}) < V_G < -0.8 \text{ V}$) where both hole-induced current and electron-induced band-to-band tunneling (BTBT) contributed to drain current (I_D). (iii) NTC region ($-2.6 \text{ V} (= V_{\text{valley}}, \text{ valley voltage}) < V_G < -1.5 \text{ V}$) where a depletion of n-type PTCDI-C13 occurred, thus $|I_D|$ was decreased with the increasing $|V_G|$, and (iv) on-state ($-5 \text{ V} < V_G < -2.6 \text{ V}$) with more hole accumulation in the higher $|V_G|$ range (Fig. 2e). A more detailed description for the origin of I_D in each regime of HTR can be found in Supplementary Fig. 8. All the HTR devices exhibited distinct NTC behaviors in the transfer characteristic.

We added detailed description of the operating principle of the HTR:

Supplementary Figure 8 | The operating principle of the HTR divided by three distinguishable regimes.

A discussion was added accordingly:

Page 10 in Supplementary Information,

The operating principle of the HTR

The HTR has a structure in which one electrode forms a contact with a stacked n-type/p-type semiconductor and the other electrode contacts a p-type semiconductor only. In the p-n junction at the channel, the band-to-band tunneling (BTBT) occurs as

the conduction band in the intrinsic region aligns with the valence band in the p-type region.^{8, 9} In the valence band of the p-type region, electrons tunnel into the conduction band of the intrinsic region and current can flow across the device.

The observed negative differential transconductance (NTC) implies the existence of the BTBT current. The operation principle can be divided into three different regions, as shown in Supplementary Fig. 8:

i) $0 \text{ V} > V_G > -1.5 \text{ V}$: Electron BTBT current occurs, and hole carriers begin to accumulate. Both the BTBT current and p-channel current contribute to I_D , exhibiting a peak current ($3.5 \times 10^{-8} \text{ A}$ at $V_G = -1.5 \text{ V}$).

ii) $-1.5 \text{ V} > V_G > -2.8 \text{ V}$: The BTBT current decreased as the n-type PTCDI-C13 is depleted by the negative gate voltage bias, resulting in the decrease of I_D with a valley current ($2 \times 10^{-9} \text{ A}$ at $V_G = -2.8 \text{ V}$).

iii) $V_G < -2.8 \text{ V}$: The n-type PTCDI-C13 is completely depleted, and only the p-channel current contributes to I_D .

The references regarding BTBT were added:

8. Yoo H., On S., Lee S. B., Cho K. & Kim J. J. Negative Transconductance Heterojunction Organic Transistors and their Application to Full-Swing Ternary Circuits. *Adv. Mater.* **31**, 1808265 (2019).

9. Nourbakhsh A., Zubair A., Dresselhaus M. S. & Palacios T. Transport properties of a MoS₂/WSe₂ heterojunction transistor and its potential for application. *Nano Lett.* **16**, 1359-1366 (2016).

Comment #5) Please provide the output curves for both HTR and organic flash memory (at both pristine and programmed status, if possible). As these output curves would help to better understand the inverter ternary output transition curves at different states. Figure 1c in ref 18 is an ideal example for reference.

Response:

We appreciate Reviewer #1 for the constructive suggestion. As Reviewer #1 suggested, we added the output curves for both HTR and organic flash memory. For the flash memory, output curves for both pristine and programmed states are added, which supports that the I_D of the flash memory was successfully modulated according to the memory programming states.

We revised the manuscript as follows:

Page 12, 2nd paragraph,

The electrical characteristics of the vertically stacked 3D T-inverter were analyzed according to the programming/erasing states of the flash memory on the 1st floor (Fig. **3d** and **e**). The I_D of the flash memory was modulated successfully by programming operation (Supplementary Fig. **15**), and with the decreasing channel conductance by charge trapping into the FG, the intermediate logic state was systematically controlled as shown in the VTC of the 3D T-inverter (Fig. **3f**). The intermediate logic value was gradually increased in accordance with the increasing V_{prg} and reached to 2.49 V with $V_{prg}=+19$ V (Fig. **3g**).

We added the output curves of the HTR and organic flash memory:

Supplementary Figure 15 | The output curves of the HTR (blue) and flash memory (red) in **a**, pristine and **b**, the programmed state ($V_{prg}=+19$ V).

Discussion was added accordingly:

Page 17 in Supplementary Information,

Output characteristics of the flash memory and HTR

The output characteristics of the HTR and flash memory with each programming state is shown in Supplementary Fig. **15**. In the flash memory, I_D decreased significantly after applying V_{prg} of +19 V compared to that observed in the pristine flash memory, which supports that the channel conductance was controlled successfully by the programming operation of the flash memory.

Comment #6) In figure 3f, the intermediate logic value window decreases notably with increasing V_{prg} , in other words, the intermediate step length at the “x” axis reduces with V_{prg} . What is the reason for this behavior? Please explain it in detail.

Response:

We appreciate Reviewer #1’s valuable comment. To investigate the intermediate logic state in the T-inverter, we analyzed the current overlap between HTR and flash memory (Supplementary Fig. **17**). At $V_{prg} = +14, +16$ V, the V_G regions where the flash memory and HTR showed similar channel conductance (I_D difference less than an order of magnitude) and corresponding V_{IN} range of the intermediate logic state was relatively wide (Supplementary Fig. **17a** and **b**). Nevertheless, the I_D of the flash memory is slightly higher than that in the NTC region of HTR in this range, which caused the V_{OUT} value (1.09 and 1.47 V at $V_{prg} = +14$ and +16 V, respectively) a bit lower than ideal value ($V_{DD}/2 \sim 2.5$ V) (Supplementary Fig. **17e** and **f**). On the other hand, at the increased V_{prg} of +19 V, I_D of the flash memory became practically identical to that of HTR in the NTC region, which resulted in the intermediate logic value of 2.49 V that is close to the ideal V_{OUT} value ($V_{DD}/2 \sim 2.5$ V, Supplementary Fig. **17d** and **h**). Therefore, the V_{IN} range exhibiting the ideal intermediate logic value in T-inverters is eventually determined by the NTC region of the HTRs, and we presented a strategy in this study that can utilize the NTC region as an ideal intermediate logic value.

We revised the manuscript by adding the detailed analysis of the transistor characteristics of the flash memory with different V_{prg} and the HTR with different V_D :

Supplementary Figure 17 | The overlapped transfer characteristics of the flash memory according to V_{prg} and the HTR with different V_D ; **a**, $V_{prg}=+14$ V, $V_{D,HTR}=-4$ V, **b**, $V_{prg}=+16$ V, $V_{D,HTR}=-3.5$ V, **c**, $V_{prg}=+18$ V, $V_{D,HTR}=-3$ V and **d**, $V_{prg}=+19$ V, $V_{D,HTR}=-2.5$ V. **e-h**, The corresponding VTC of the T-inverter.

A discussion was added accordingly:

Page 13, 1st paragraph,

The intermediate logic value was gradually increased in accordance with the increasing V_{prg} and reached to 2.49 V with $V_{prg}=+19$ V (Fig. **3g**). Therefore, the optimum programming state of the flash memory allowed that the intermediate logic value is practically identical to $V_{DD}/2$ (logic "1"), which made the intermediate logic state clearly distinguishable from V_{DD} (logic "2") and G_{ND} (logic "0"). The V_{IN} range where the intermediate logic states appeared was gradually shifted with the increasing V_{prg} , because the NTC region of the HTR was shifted to lower $|V_G|$ region with the decreasing $|V_D|$ (Supplementary Fig. **16**). The V_{IN} range of the intermediate

logic states decreased slightly with the increasing V_{prg} , because only the NTC region in the HTR could be represented to the intermediate logic state, which enabled us to obtain the ideal V_{out} value ($V_{\text{DD}}/2$) by optimizing the memory programming process (Supplementary Fig. 17). The 1st gain decreased gradually while 2nd gain increased with the increasing V_{prg} , resulting from the increased intermediate logic value (Fig. 3h and Supplementary Fig. 18).

Page 19 in Supplementary Information,

The overlapped transfer curves of the transistors and resulting VTC of the T-inverter according to memory programming state

Supplementary Fig. 17a-d shows the overlapped transfer curves of the flash memory and HTR to analyze the intermediate logic state according to the memory programming state. The VTCs of the T-inverter was also shown in Supplementary Fig. 17e-h along with each memory programming state. With the low V_{prg} (+14 and +16 V), the V_{G} region where the flash memory and HTR exhibited similar I_{D} (less than an order of magnitude difference) was relatively wide (~ 1.2 V) (Supplementary Fig. 17a, b). In those V_{G} regions, the flash memory showed slightly higher I_{D} compared to that of the HTR, resulting in low intermediate logic value (1.09 and 1.47 V at $V_{\text{prg}} = +14$ and +16 V, respectively). However, with the V_{prg} of +19 V, both transistors showed quite similar I_{D} values in the NTC of the HTR, which produced the ideal V_{OUT} value ($\sim V_{\text{DD}}/2$) in the intermediate logic state. In this optimum programming state, the V_{IN} range of the intermediate logic state decreased slightly to ~ 0.9 V, because only the NTC region in the HTR can represent the intermediate logic state.

Comment #7) It is not convincing to use extrapolation to describe the retention time in Figure 4b. Moreover, the authors only show the I_{a} , intermediate logic values, and gain values at 10,000 s, which is relatively short. Please extend the retention time to 100,000 s (approximately 28 hours), which is doable in general (*Adv. Mater.* **2015**, *27*, 6257-6264, *Nat. Mater.* **2008**, *7*, 547-550.).

Response:

We appreciate Reviewer #1's constructive comment. As Reviewer #1 recommended, we measured the electrical characteristics of the devices with the extended retention time up to 100,000 s. However, the $I_{10,000s}/I_{0s}$ values on the y-axis in Fig. 4c was used instead of $I_{100,000s}/I_{0s}$, to accommodate the previous studies as many as possible for comparison of the retention performance, because $I_{10,000s}$ rather than $I_{100,000s}$ information was presented in most of the references. Instead, we added a summary of the retention characteristics of the flash memory developed in this study compared to previously reported organic floating-gate flash memories adapting polymer dielectrics, by using both $I_{10,000s}/I_{0s}$ and $I_{100,000s}/I_{0s}$ (Supplementary Table 3).

We fully agree with Reviewer #1 that the long-term retention by extrapolation is not an experimental value but a predicted one, even though we tried our best to extract as strictly as possible. Therefore, we deleted long-term retention by extrapolation, and revised x-axis in Figure 4b so that the measured value could be seen clearly.

We revised the manuscript by adding the change in the electrical characteristics of the devices with the prolonged time:

Figure 4 | Stability of the 3D T-inverter. **a**, The change in transfer characteristics of the flash memory according to time after the optimum programming state ($V_{\text{prg}}=+19$ V) and **b**, I_D at $V_G = 1$ V and $I_D = 2$ V versus time in the programming (open symbol) and erasing (closed symbol) states. **c**, The comparison of the retention characteristics and $V_{\text{prg}}/V_{\text{ers}}$ of the flash memory in this study to those based on polymer dielectrics in the previous reports. **d**, The change in transfer curves of the flash memory according to time and the transfer curve of the HTR. **e**, The change in VTCs and **f**, intermediate logic value of the 3D T-inverter according to time.

Supplementary Figure 22 | **a**, The change in transfer curves of the flash memory, **b**, VTCs and **c**, DC gain profile of the 3D T-inverter according to time. **d**, The intermediate logic value, **e**, 1st and 2nd gain values versus time extracted from VTCs.

Supplementary Figure 23 | The DC gain profiles of the 3D T-inverter with the optimum memory programming state ($V_{prg}=+19$ V) according to time.

We revised the manuscript accordingly:

Page 15, 2nd paragraph,

To investigate the memory retention, the transfer curves of the optimum programming state ($V_{\text{prg}}=+19$ V) were monitored with time (Fig. **4a**). The flash memory showed negligible change in transfer characteristics over time with only the **0.7 V** of ΔV_T even after **10^5 s**. The change in I_D at $V_G=1$ V and $V_D=2.5$ V was extracted from the transfer curves with the programming/erasing states (Fig. **4b** and Supplementary Fig. **22**). The I_D in on-state (erasing state) at **10^3 , 10^4 and 10^5 s** showed **0.999, 0.911 and 0.716 times** of the initial I_D , respectively, **which indicates the I_D change less than an order of magnitude even after 10^5 s**. Such excellent retention characteristics enabled the flash memory to retain sufficient I_D difference in the programming/erasing states over time. To the best of our knowledge, the flash memory device developed in this study exhibited one of the best retention characteristics as well as lowest $V_{\text{prg}}/V_{\text{ers}}$ among the organic FG flash memory based on polymer dielectrics, resulting from the tunneling-based programming mechanism through the robust dielectric layers with high α_{CR} (Fig. **4c** and **Supplementary Table 3**).⁶⁰⁻⁷⁰

Page 16, 1st paragraph,

Accordingly, the 3D T-inverter fully maintained its three distinct logic states with full-swing operation over time owing to the excellent retention performance of the flash memory (Fig. **4d** and **4e**). Also, no notable device degradation was observed, such as hysteresis behavior. **The intermediate logic values were kept close to the ideal value (2.45, 2.47 and 2.42 V at 10^3 , 3×10^3 and 10^4 s, respectively) and became 2.18 V even after 10^5 s (Fig. **4f**).** The intermediate logic value was slightly decreased with time, **because the transfer curve of the flash memory was shifted to negative V_G direction, which made I_D of the flash memory higher than that of the HTR in the NTC region (Fig. **4d**).** Because of the excellent retention of three evidently discernable logic states, **high DC gain values of the measured 3D T-inverter were maintained up to 10^5 s (Supplementary Fig. **23**).** The 1st and 2nd gain values were also retained to **-13.6 and -20.8 V/V, respectively, and their change was less than 4.5 V/V for the 1st gain and 3.0 V/V for the second gain over the whole range of the measuring time.** The 3D T-

inverter with erasing state of the flash memory also maintained its electrical characteristics, even though the full-swing operation could not be retained. The changes in the intermediate logic value and DC gain value less than 0.10 V and 5.5 V/V, respectively, even after 10^5 s (Supplementary Fig. 22). It follows from the observations above that the electrical characteristics of the 3D T-inverter were fully preserved once the constituent flash memory was adequately programmed, mainly due to the excellent retention performance of the flash memory developed in this study.

Page 27 in Supplementary Information,

The retention characteristics of the flash memory in erasing operation

The retention characteristics in memory erasing operation were also investigated. The change in transfer characteristics with time after applying the maximum V_{ers} (-8 V) are shown in Supplementary Fig. 22a, which showed negligible change over time with only 0.6 V of ΔV_T after 10^5 s. This excellent retention performance led little change in the electrical characteristics of the 3D T-inverter over time (Supplementary Fig. 22b-c). Also, there was no notable device degradation such as hysteresis behavior. The change in intermediate logic value was less than 0.10 V even after 10^5 s (Supplementary Fig. 22d). Even with the extremely small change in the transfer curves, V_{OUT} at $V_{\text{IN}}=0$ V was recovered from 4.45 to 4.78 V in the VTC after 10^5 s, which induced the enhanced voltage swing and the improved 1st gain value from -17.7 to -23.1 V/V (Supplementary Fig. 22e). Nevertheless, there was still negligible change in the 2nd gain value (less than 0.5 V/V) throughout the whole measurement time.

Page 29 in Supplementary Information,

The DC gain profiles of the flash memory in programming state over time

The changes in DC gain profiles of the flash memory with the optimum programming state ($V_{\text{prg}}=+19$ V) are shown in Supplementary Fig. 23. Only negligible change was

observed in the DC gain values (less than 4.5 and 3.0 V/V for the 1st and 2nd gain values, respectively) and their V_{IN} positions (less than 0.35 V) even after 10^5 s.

We added the table that summarizes the retention characteristics of the organic floating-gate flash memory:

Page 28 in Supplementary Information,

Comparison of the retention characteristics

Supplementary Table 3 summarizes the retention characteristics of the flash memory developed in this study compared to the previously reported organic flash memories employing polymer dielectric materials.

Supplementary Table 3 | Summary of the retention characteristics of the reported floating-gate flash memory based on polymer dielectric.

Year ^[reference]	V_{prg}, V_{prg} (V)	$I_{10,000s}/I_{0s}$	$I_{100,000s}/I_{0s}$
2009 ^[60]	30	0.544	N/A
2010 ^[61]	90	0.448	0.316
2013 ^[62]	70	0.291	N/A
	70	0.126	
2014 ^[63]	6	0.697	0.612
	50	0.552	
2015 ^[64]	50	0.485	N/A
	50	0.035	
2015 ^[65]	50	0.428	N/A
2015 ^[66]	80	0.637	0.386
	150	0.175	
2016 ^[67]	150	0.119	N/A
	150	0.109	
	150	0.059	
2018 ^[68]	60	0.149	N/A
2018 ^[69]	80	0.303	N/A
	80	0.071	

	80	0.010	
2021 ^[70]	30	0.599	N/A
	30	0.528	
This work	19	0.911	0.716

Comment #8) The author claimed an excellent retention characteristic on pages 14-15 based on the fact that the I_d maintained 0.911 times of the initial I_d at 10,000s. However, to our knowledge, this nearly 10% drop in I_d in less than 3 hours is a sign of moderate retention property (*ACS Appl. Mater. Interfaces* **2018**, *10*, 9563-9570; *Nano Lett.* **2009**, *9*, 1713-1719. *Nat Comm.* **2014**, *5*, 4720.). Further, the extrapolation method is not strictly scientific.

Response:

We appreciate Reviewer #1's constructive comment. However, we still believe that the organic flash memory developed in this study showed excellent retention performance. As shown in **Table R5**, the I_D change was less than an order of magnitude throughout measuring time, which is clearly one of the highest retention performances compared to the previously developed organic floating-gate flash memories employing polymer dielectric materials (Fig. **4c** and Supplementary Table **3**).

Table R5. The I_D values measured at $V_G=1$ V according to time.

Time (s)	0	50	100	300	1,000	3,000	10,000	100,000
I_D at $V_G=1$ V ($\times 10^{-7}$ A)	1.347	1.351	1.352	1.355	1.345	1.264	1.228	0.965

We also carefully checked the retention performance in the previous reports that Reviewer #1 mentioned. Also, we compared our retention performance to other organic memories based on charge trapping mechanism, which are summarized in **Table R6**.

Table R6. Summary of the operating voltage and DC gain values in this study compared to the previous ternary logic circuits

Reference	Memory operating principle	V_{prg} or V_{ers} (V)	$I_{10,000\text{s}}/I_{0\text{s}}$	$I_{100,000\text{s}}/I_{0\text{s}}$
Nat. Mater. 2008 , 7, 547-550	Ferroelectric	20	0.768	0.632
ACS Appl. Mater. Interfaces 2018 , 10, 9563-9570	Electrical double-layer	9	0.334	0.492
Nat Commun. 2014 , 5, 4720.	Photonic flash memory	40	0.605	0.249
Adv. Mater. 2015 , 27, 6257-6264	Charge trapping	50	0.912	0.639
Adv. Funct. Mater. 2020 , 30, 2004665	Charge trapping	16	0.698	0.576
ACS Appl. Mater. Interfaces 2015 , 7, 10957-10965	Charge trapping	30	0.454	N/A
Chem. Asian. J. 2016 , 11, 1631-1640	Charge trapping	40	0.394	N/A
Adv. Electron. Mater. 2019 , 5, 1800799	Charge trapping	14	0.340	0.078
This work	Flash memory	19	0.911	0.716

As summarized in **Table R6**, our flash memory showed the excellent retention performance compared to the values reported previously. This performance is even comparable to those observed in the organic charge trapping memories, which typically showed better retention performance compared to organic floating-gate flash memories.

We revised the manuscript by as follows:

Page 16, 1st paragraph,

To the best of our knowledge, the flash memory device developed in this study exhibited one of the best retention characteristics as well as lowest $V_{\text{prg}}/V_{\text{ers}}$ among the organic FG flash memory based on polymer dielectrics, resulting from the tunneling-based programming mechanism through the robust dielectric layers with

high α_{CR} (Fig. **4c** and Supplementary Table **3**).⁶⁰⁻⁷⁰ Moreover, the flash memory showed the retention characteristics even superior or at least comparable to those obtained from the organic memories with different operating principles including charge trapping memories and photonic memories.⁷¹⁻⁷⁸ Accordingly, the 3D T-inverter fully maintained its three distinct logic states with full-swing operation over time owing to the excellent retention performance of the flash memory (Fig. **4d** and 4e).

71. Asadi K., De Leeuw D. M., De Boer B. & Blom P. W. Organic non-volatile memories from ferroelectric phase-separated blends. *Nat. Mater.* **7**, 547-550 (2008).

72. Koo J. et al. Nonvolatile electric double-layer transistor memory devices embedded with Au nanoparticles. *ACS Appl. Mater. & Interfaces* **10**, 9563-9570 (2018).

73. Zhou Y. et al. An upconverted photonic nonvolatile memory. *Nat. Commun.* **5**, 4720 (2014).

74. Chiu Y. C. et al. Oligosaccharide carbohydrate dielectrics toward high-performance non-volatile transistor memory devices. *Adv. Mater.* **27**, 6257-6264 (2015).

75. Wang W. et al. Highly reliable top-gated thin-film transistor memory with semiconducting, tunneling, charge-trapping, and blocking layers all of flexible polymers. *ACS Appl. Mater. & Interfaces* **7**, 10957-10965 (2015).

76. Tung W. Y. et al. High performance nonvolatile transistor memories utilizing functional polyimide-based supramolecular electrets. *Chem. Asian J.* **11**, 1631-1640 (2016).

77. Pak K., Choi J., Lee C. & Im S. G. Low-power, flexible nonvolatile organic transistor memory based on an ultrathin bilayer dielectric stack. *Adv. Electron. Mater.* **5**, 1800799 (2019).

78. Lee C. et al. Long-term retention of low-power, nonvolatile organic transistor memory based on ultrathin, trilayered dielectric containing charge trapping functionality. *Adv. Funct. Mater.* **30**, 2004665 (2020).

Reviewer (#2)'s COMMENTS:

In the manuscript, multi-valued logic inverter based on a heterojunction transistor and a nonvolatile floating-gate transistor were constructed with one heterojunction transistor stacked vertically on top of another nonvolatile floating-gate transistor. Intermediate V_{out} value of $\sim V_{DD}/2$, gain as high as 20 V/V were achieved, along with low driving voltage of less than 5 V. Moreover, long-term stability of the inverter was also obtained. This work is interesting and novel. It can be published in *Nat. Comm.* if the following comments can be properly addressed.

Response:

We appreciate the Reviewer #2's encouraging and constructive comments. We present the point-by-point response for each comment as follows.

Comment #1) In the experimental section, details for the inverter fabrication are missing, such as the thicknesses of the floating gate electrode and other relevant layers, and layer dimensions. These parameters are extremely important for others to reproduce the work.

Response:

We appreciate Reviewer #2's valuable comment. As Reviewer #2 suggested, we revised '**Device fabrication & characterization**' in Method section by adding the detailed information. Also, we corrected the channel dimension (800 μm of the width and 400 μm of the length).

We revised the manuscript by as follows:

Page 27, in Device fabrication & characterization section,

Device fabrication & characterization

To fabricate the vertically stacked organic 3D T-inverter, the flash memory device was fabricated. For gate and FG electrode, Al was thermally evaporated with the deposition rate of 0.1 nm/s. The thickness of the gate and FG electrode was commonly set to 50 nm, which was monitored *in-situ* by quartz crystal microbalance (QCM), and all the thermal evaporation processes were performed under high vacuum lower than 2×10^{-6} torr. The dielectric layers were deposited via iCVD process as

described above and the thicknesses of the pC1D1 BDL and pV3D3 TDL were 100 and 24 nm, respectively. The organic semiconductors including DNNT and PTCDI-C13 were also thermally deposited with the deposition rate of 0.03 nm/s. The PTCDI-C13 was recrystallized by the following thermal annealing at 200 °C for an hour. For *S/D* electrodes, 70 nm-thick Au was thermally evaporated with the deposition rate of 0.1 nm/s. The channel width and length were 800 and 400 μm, respectively. 1 μm-thick pV3D3 was deposited onto the flash memory by the iCVD process as an ILD, and the HTR was fabricated directly on top of the ILD. For the fabrication of the HTR, 50 nm-thick Al electrode was thermally evaporated and used as gate electrode and the thickness of pV3D3 dielectric layer was 40 nm. The partial PTCDI-C13 semiconducting layer was deposited and recrystallized at 200 °C for an hour, followed by the DNNT layer that ranges from *S* to *D* electrode. 70 nm-thick Au *S/D* electrode was deposited and the *D* electrode of the HTR was connected to *D* electrode of the flash memory through via-hole-less multi-metal interconnection. The overall areal dimension of the active area was 0.64 mm² (0.8 mm by 0.8 mm for width and length) including *S/D* electrodes, and the areal dimension of the patterned dielectric layer was 1.44 mm² (1.2 mm by 1.2 mm for width and length). To visualize the 3D T-inverter, the device was sliced by a focused ion beam (Helios Nanolab 450) and cross-sectional HRTEM images were taken by Cs-corrected TEM (Titan cubed G2, FEI) with EDS mapping analysis.

Comment #2) It is mentioned that a well-defined intermediate logic V_{out} state is important. How about the intermediate V_{in} value in T-inverters? Is the current work showing satisfactory intermediate V_{in} ?

Response:

We appreciate Reviewer #2's valuable comments. As Reviewer #2 pointed out, the V_{IN} value that corresponds to the intermediate logic state is also an important factor to determine the T-inverter performance.

In our device, the intermediate logic states were located at V_{IN} of 2.5 V with the pristine state and low V_{prg} including +14 and +16 V, however, the intermediate logic state was shifted toward higher V_{IN} range. This is because the NTC region was shifted to lower gate voltage ($|V_G|$) region with the decreasing $|V_D|$ as illustrated in Supplementary Fig. **16**, which is mainly resulting from the reduced electron injection with the decreasing gate-to-drain voltage (V_{GD}). Therefore, with the optimum V_{prg} (+19 V), the intermediate logic value was shifted with the decreased $|V_D|$ of the HTR. In this regard, the optimum programming operation of the flash memory can force the I_D of the flash memory to match that of the HTR in the NTC region, which enabled the ideal V_{OUT} amplitude value (Supplementary Fig. **17**). This ideal intermediate logic V_{IN} value could be achieved by the proposed strategy in this report where the flash memory was integrated into the T-inverter for the first time. In other words, the range of the intermediate logic V_{IN} state is determined mainly by the NTC region of a HTR, and we presented an effective way in this study to utilize the NTC behavior as an ideal V_{OUT} value in the intermediate logic state.

The length and position of the NTC region in the HTR directly determines the V_{IN} range and position of the intermediate logic state in the T-inverter, respectively. Therefore, systematic control of the length and position of NTC region in the HTR is highly required to achieve high-performance T-inverter. To achieve this goal, we are currently investigating two different strategies: (i) improving the charge transport characteristics of a n-type semiconductor to enhance the length of NTC region, and (ii) introducing an additional dielectric layer that can induce threshold voltage (V_T) shift to control the position of the NTC region (**Fig. R1**).

Figure R1. Schematic illustration of the change in transfer characteristics of a HTR according to the **a**, improving n-type semiconductor performance and **b**, introducing V_T controlling layer.

As a preliminary experiment to verify the validity of the strategy (i), we increased the thickness of n-type PTCDI-C13 partial semiconductor ($d_{PTCDI-C13}$) to facilitate the free carrier density and charge transport (*Appl. Phys. Lett.* **2011**, 98, 073307 and *Org. Electron.* **2010**, 11, 1920-1927). With the increasing thickness of n-type semiconductor, the length of NTC region was improved from 0.8 to 1.1 V (**Fig. R2**), which showed feasibility of the strategy (i).

Figure R2. The device structure of the HTR employing n-type PTCDI-C13 semiconductor with thickness of **a**, 30 and **b**, 45 nm. **c**, The transfer characteristics of the HTR according to the PTCDI-C13 thickness and **d**, V_{peak} , V_{valley} extracted from the transfer curves.

Also, we introduced the V_T control layer to verify that the position of NTC region can be controlled (strategy (ii)). In this experiment, the thicknesses of both semiconductors were commonly fixed to 30 nm. As shown in **Fig. R3a-d**, the NTC region was shifted toward higher $|V_G|$ after employing the V_T control layer without hampering the length of NTC region. Moreover, the ideal V_{OUT} value in the intermediate logic state was achieved at the V_{IN} of 2.5 V (**Fig. R3e-f**). Please note that the flash memory with the optimum programming state ($V_{prg}=+17$ V) was still very necessary to achieve the I_D balance between HTR and a complementary transistor (**Fig. R3e**).

Figure R3. The device structure of **a**, control HTR and **b**, HTR with a V_T control layer. **c**, The transfer characteristics of the HTR with respect to the presence of V_T control layer and **d**, V_{peak} , V_{valley} extracted from the transfer curves. **e**, the overlapped transfer curves of HTR employing

the V_T control layer and flash memory with the optimum programming state ($V_{\text{prg}}=+17$ V). **f**, The voltage transfer curve (VTC) of the T-inverter composed of the HTR employing V_T control layer and flash memory with the optimum programming state ($V_{\text{prg}}=+17$ V).

Even though we could accomplish the matching between V_{IN} and V_{OUT} values in the intermediate logic state in our preliminary experimental results as shown above, it is still highly required to enlarge the V_{IN} range where the intermediate V_{OUT} value appeared, to develop the high-performance T-inverter with the maximized noise margin. To achieve this goal, the position of the NTC region should further be optimized and its range should be enlarged by combining the strategy (i) and (ii) described above. As we verified that the length of NTC region can be increased with the improved electrical characteristics of n-type partial semiconductor (**Fig. R2**), it can be replaced to other high-performance n-type semiconductor such as two-dimensional molybdenum disulfide (2D MoS₂) and amorphous indium-gallium-zinc oxide (a-IGZO) for further improvement. Also, employing functional dielectric layer that can control the V_T of n-type OTFT or high- k dielectric layers developed in our previous reports (*Adv. Funct. Mater.* **2016**, *26*, 6574-6582, *ACS Appl. Mater. Interfaces* **2017**, *9*, 20808-20817 and *Adv. Electron. Mater.* **2020**, *6*, 200314) can allow us to control the position of the NTC region.

Together with the optimization of the NTC behavior of the HTR, it is of critical importance to introduce the floating-gate memory structure into the complementary transistor as we demonstrated in this study, in order to match the channel conductance of the HTR and complementary transistor and to utilize the NTC region of the HTR as an ideal intermediate logic state. Considering all the aspects described above, we are indeed currently working on the optimization of the NTC behavior of the HTR and corresponding T-inverter performance with appropriate design/simulation approach. However, we believe these strategies to control of NTC behavior are beyond the scope of this report, and we hope that we can show more convincing result in near future as a separate report with systematic study.

We revised the manuscript by adding the transistor characteristics of HTR with different V_D and the overlapped transfer curves of the flash memory and HTR to analyze the T-inverter characteristics:

Supplementary Figure 16 | **a**, The transfer characteristics of the HTR with different V_D . **b**, the change in V_{peak} and V_{valley} with respect to V_D .

Supplementary Figure 17 | The overlapped transfer characteristics of the flash memory according to V_{prg} and the HTR with different V_D ; **a**, $V_{\text{prg}} = +14$ V, $V_{D,\text{HTR}} = -4$ V, **b**, $V_{\text{prg}} = +16$ V, $V_{D,\text{HTR}} = -3.5$ V, **c**, $V_{\text{prg}} = +18$ V, $V_{D,\text{HTR}} = -3$ V and **d**, $V_{\text{prg}} = +19$ V, $V_{D,\text{HTR}} = -2.5$ V. **e-h**, The corresponding VTC of the T-inverter.

A discussion was added accordingly:

Page 13, 1st paragraph,

The intermediate logic value was gradually increased in accordance with the increasing V_{prg} and reached to 2.49 V with $V_{\text{prg}}=+19$ V (Fig. **3g**). Therefore, the optimum programming state of the flash memory allowed that the intermediate logic value is practically identical to $V_{\text{DD}}/2$ (logic "1"), which made the intermediate logic state clearly distinguishable from V_{DD} (logic "2") and G_{ND} (logic "0"). The V_{IN} range where the intermediate logic states appeared was gradually shifted with the increasing V_{prg} , because the NTC region of the HTR was shifted to lower $|V_{\text{G}}|$ region with the decreasing $|V_{\text{D}}|$ (Supplementary Fig. **16**). The V_{IN} range of the intermediate logic states decreased slightly with the increasing V_{prg} , because only the NTC region in the HTR could be represented to the intermediate logic state, which enabled us to obtain the ideal V_{out} value ($V_{\text{DD}}/2$) by optimizing the memory programming process (Supplementary Fig. **17**). The 1st gain decreased gradually while 2nd gain increased with the increasing V_{prg} , resulting from the increased intermediate logic value (Fig. **3h** and Supplementary Fig. **18**).

Page 18 in Supplementary Information,

The transfer characteristics of the HTR with different V_{D}

To investigate the origin that causes the shift of the intermediate logic state, we measured the transfer characteristics of the HTR with different V_{D} , because V_{D} of the HTR is determined by the difference between the supply voltage and the intermediate logic value (Supplementary Fig. **16a**). With the decreasing $|V_{\text{D}}|$ of the HTR, the NTC region was shifted toward lower $|V_{\text{G}}|$, which led the shift of the intermediate logic state to higher V_{IN} . Nevertheless, the length of the NTC region was fully preserved (Supplementary Fig. **16b**) The shift of the NTC region is most likely resulting from the reduced electron injection with the decreasing gate-to-drain voltage (V_{GD}) so that the depletion of n-type PTCDI-C13 started to occur at lower $|V_{\text{G}}|$.

Page 19 in Supplementary Information,

The overlapped transfer curves of the transistors and resulting VTC of the T-inverter according to memory programming state

Supplementary Fig. **17a-d** shows the overlapped transfer curves of the flash memory and HTR to analyze the intermediate logic state according to the memory programming state. The VTCs of the T-inverter was also shown in Supplementary Fig. **17e-h** along with each memory programming state. With the low V_{prg} (+14 and +16 V), the V_G region where the flash memory and HTR exhibited similar I_D (less than an order of magnitude difference) was relatively wide (~ 1.2 V) (Supplementary Fig. **17a, b**). In those V_G regions, the flash memory showed slightly higher I_D compared to that of the HTR, resulting in low intermediate logic value (1.09 and 1.47 V at $V_{prg} = +14$ and +16 V, respectively). However, with the V_{prg} of +19 V, both transistors showed quite similar I_D values in the NTC of the HTR, which produced the ideal V_{OUT} value ($\sim V_{DD}/2$) in the intermediate logic state. In this optimum programming state, the V_{IN} range of the intermediate logic state decreased slightly to ~ 0.9 V, because only the NTC region in the HTR can represent the intermediate logic state.

Comment #3) The hysteresis of a transistor and inverter are highly dependent on the sweeping speed of the input voltage. It is not accurate to say this transistor/inverter is hysteresis-free without mentioning the sweeping speeds.

Response:

We appreciate Reviewer #2 for the constructive comment. We measured the transfer characteristics of the transistors and VTC of the inverter with different sweeping speed, which verified that no notable hysteresis behavior was obtained regardless of the sweeping speed ranging from 0.47 to 0.03 V/s, as shown in Supplementary Fig. **10** and **11**. We added the sweeping speed information in the revised manuscript.

We revised the manuscript by as follows:

Page 11, 2nd paragraph,

All the fabricated 3D T-inverters showed full-swing operation with the uniform electrical characteristics stemming from the excellent device-to-device uniformity of the constituent devices, together with the high I_{on}/I_{off} ratio of the HTR, which can be hardly achieved in conventional AATs where two semiconductors are overlapped only at the center of the channel.^{27, 28} Also, no notable hysteresis was observed in the transfer curves of the flash memory and HTR, and the VTC of the T-inverter regardless of the sweeping speed (Supplementary Fig. **10** and **11**). Moreover, all the 3D T-inverters displayed a distinct intermediate logic state with two clearly distinguishable maximum gain values.

Page 28, in **Device fabrication & characterization section**,

The electrical characteristics were measured by B1500A semiconductor analyzer (Agilent Technologies). The sweeping speed of the input voltage was set to ~ 0.1 V/s, otherwise specified. The temperature of the devices was measured by infrared (IR) thermometer (Fluke 62 MAX). All the device fabrication and characterization were performed in the N₂-filled glovebox.

We added transfer curves of the transistors and VTC of the inverter with different sweeping speed.

Supplementary Figure 10 | The transfer curves of the **a**, flash memory and **b**, HTR with different sweeping speed.

Supplementary Figure 11 | The VTCs of the T-inverter with different sweeping speed.

Discussion was added accordingly:

Page 12 in Supplementary Information,

The electrical characteristics of the transistors and inverter with different measuring speed.

The transfer characteristics of the flash memory and HTR measured with each gate voltage (V_G) sweeping speed were shown in Supplementary Fig. **10**. The VTCs of the T-inverter with each input voltage (V_{IN}) sweeping speed were also shown in Supplementary Fig. **11**. No notable hysteresis was observed, and extremely low gate leakage current was fully maintained regardless of the sweeping speed in the transfer curves of the transistors. The T-inverter also showed only negligible amount of hysteresis with the sweeping speed as low as 0.06 V/s.

Comment #4) Related to the last comments, how fast the inverter can be switched? What is the delay time? This is one important parameter for an inverter.

Response:

We appreciate Reviewer #2 for the constructive comment. As Reviewer #2 suggested, we measured the switching speed values of the T-inverter. A switching speed within a few hundreds of milliseconds was obtained, which is comparable to those obtained generally in the organic binary logic circuits, which showed that the switching speed was not degraded with the increasing number of logic states. The switching speed of the OTFT is closely related to the channel length, charge mobility and contact resistance (*Org. Electron.* **2017**, *49*, 179-186, *Adv. Funct. Mater.* **2020**, *30*, 1909501 and *Sci. Adv.* **2020**, *6*, eaaz5156). Therefore, the switching speed of our device can further be improved by optimizing the channel dimension and fabrication process.

We revised the manuscript by adding switching speed information:

Supplementary Figure 21 | **a**, The applied V_{IN} and **b**, V_{OUT} in response to the applied V_{IN} of the T-inverter with the optimum programming state.

We also added the summary of the measured switching speed:

Supplementary Table 2 | Summary of the parameters for the switching speed of the T-inverter.

Switching number	V_{OUT} (V)		Time (s)		Switching speed (ms)
	initial	terminal	initial	terminal	
1	0	2.6	0.60	0.82	220
2	2.6	5	1.19	1.31	120
3	5	2.6	1.80	2.09	290
4	0	5	2.99	3.12	130
5	5	0	3.60	3.91	310
6	0	2.6	4.19	4.41	220
7	2.6	5	4.81	5.12	310

Discussion was added accordingly:

Page 14, 2nd paragraph,

The 3D T-inverter with both states exhibited no notable hysteresis regardless of the V_{IN} history throughout the measuring time. However, the intermediate logic value was clearly enhanced in the programming state, compared to that observed in the pristine state. The switching speed of the T-inverter was also investigated (Supplementary Fig. 21 and Supplementary Table 2), which showed that the switching speed was comparable to those obtained frequently in the organic binary logic circuits.⁵⁷⁻⁵⁹ It follows from the above observation that the electrical characteristics of the vertically stacked 3D T-inverter, particularly the intermediate logic state, could be controlled systematically by the programming/erasing states of the flash memory.

57. Zschieschang U., Bader V. P. & Klauk H. Below-one-volt organic thin-film transistors with large on/off current ratios. *Org. Electron.* **49**, 179-186 (2017).

58. Chang J. S., Facchetti A. F. & Reuss R. A circuits and systems perspective of organic/printed electronics: review, challenges, and contemporary and emerging design approaches. *IEEE J. Emerg. Sel. Top.* **7**, 7-26 (2017).

59. Kumar B., Kaushik B. K. & Negi Y. S. Organic thin film transistors: structures, models, materials, fabrication, and applications: a review. *Polym. Rev.* **54**, 33-111 (2014).

Page 25 in Supplementary Information,

The analysis on the switching speed of the T-inverter

To investigate the switching speed of the 3D T-inverter with the optimum programming state, we measured the V_{OUT} in the response to the applied V_{IN} according to time (Supplementary Fig. 21 and Supplementary Table 2). The switching speed of the T-inverter was calculated as the difference between the time when a 10% increase in V_{OUT} occurs and the time when a 90% of the terminal V_{OUT} was achieved. A switching speed within a few hundreds of milliseconds was obtained, which is comparable to those obtained generally in the organic binary logic circuits.¹⁰⁻¹²

10. Zschieschang U., Bader V. P. & Klauk H. Below-one-volt organic thin-film transistors with large on/off current ratios. *Org. Electron.* **49**, 179-186 (2017).
11. Chang J. S., Facchetti A. F. & Reuss R. A circuits and systems perspective of organic/printed electronics: review, challenges, and contemporary and emerging design approaches. *IEEE J. Emerg. Sel. Top.* **7**, 7-26 (2017).
12. Kumar B., Kaushik B. K. & Negi Y. S. Organic thin film transistors: structures, models, materials, fabrication, and applications: a review. *Polym. Rev.* **54**, 33-111 (2014).

Comment #5) It is suggested to plot/draw the illustration of the inverter/transistor with both side view and top view, so that the structure of the devices can be much clearer.

Response:

We appreciate Reviewer #2 for the valuable comment. As Reviewer #2 suggested, we have added the 3D images to clearly show the structure of the devices.

We revised the manuscript by adding the 3D images of the device:

Supplementary Figure 1 | **a**, A schematic of the device and **b**, Microscopic image and schematic illustration of the device with different angles according to the fabrication procedure (top: organic flash memory in the 1st floor, bottom: organic HTR in the 2nd floor).

Discussion was added accordingly:

Page 6, 2nd paragraph,

The schematic illustration for the fabrication process and the structure of the vertically stacked organic ternary logic inverter are shown in Fig. **1a** and **b**, respectively. All the dielectric layers were deposited by iCVD process and patterned *in-situ* through shadow mask to achieve via-hole-less metal interconnection.¹⁶ Also, a 1 μm -thick interlayer dielectric (ILD) was imposed between the unit devices to electrically isolate the flash memory and HTR. Schematic illustration of the device structure with various view points according to the fabrication process is shown in Supplementary Fig. **1**. Organic flash memory including FG was introduced to match the channel conductance

with the HTR in the NTC region, thus to optimize the voltage transfer characteristic (VTC) and intermediate logic state of the resulting 3D T-inverter (Fig. **1c** and **d**).

Page 2 in Supplementary Information,

Schematic illustration of the device

The schematic illustration of the device is shown in Supplementary Fig. **1**. Note that the gate electrode of the flash memory and HTR was connected and the drain electrode of the flash memory and HTR was connected through the via-hole-less metal interconnection (Supplementary Fig. **1a**). Supplementary Fig. **1b** shows the optical microscopy image of the fabricated device and corresponding schematic illustration of the device according to the fabrication procedure.

Reviewer (#3)'s COMMENTS:

Excellent paper, very well structured, novel and interesting. The authors incorporate several novel ideas, including the use of new CVD-deposited insulators, and the use of these materials to create a flash memory. The paper further includes a comprehensive characterization of the fabricated devices and uses more than enough devices and sweep measurements to confirm a scalable and reliable process. The paper also includes a reasonable background and cites the background work of other groups well.

I recommend for acceptance without any major modifications necessary.

Response:

We appreciate the positive and encouraging comments from Reviewer #3.

We very appreciate the Reviewers' all the valuable comments, which greatly helped us to improve this manuscript.

Response to the Reviewers' Comments

The authors thank the Reviewers for their considerate review of our manuscript and the valuable comments.

Reviewer (#1)'s COMMENTS:

The author had addressed most comments from the first round review. It can be published now in *Nature Communications*.

Response:

We appreciate the Reviewer for the valuable comments.

Reviewer (#2)'s COMMENTS:

The authors took into consideration my comments and addressed them in full. Thus, I believe that the paper can be published as it is.

Response:

We appreciate the Reviewer for the valuable comments.